# A Short History of the First 50 Years: From the GRB Prompt Emission and Afterglow Discoveries to the Multimessenger Era

Filippo Frontera [1,2]

1   Department of Physics and Earth Science, University of Ferrara, 44122 Ferrara, Italy; frontera@fe.infn.it
2   Italian National Institute of Astrophysics (INAF), Astrophysics and Space Science Observatory (OAS), Via Gobetti 101, 40129 Bologna, Italy

**Abstract:** More than fifty years have elapsed from the first discovery of gamma-ray bursts (GRBs) with American *Vela* satellites, and more than twenty-five years from the discovery with the *BeppoSAX* satellite of the first X-ray afterglow of a GRB. Thanks to the afterglow discovery and to the possibility given to the optical and radio astronomers to discover the GRB optical counterparts, the long-time mystery about the origin of these events has been solved. Now we know that GRBs are huge explosions, mainly ultra relativistic jets, in galaxies at cosmological distances. Starting from the first GRB detection with the Vela satellites, I will review the story of these discoveries, those obtained with *BeppoSAX*, the contribution to GRBs by other satellites and ground experiments, among them being *Venera*, *Compton Gamma Ray Observatory*, *HETE-2*, *Swift*, *Fermi*, *AGILE*, *MAGIC*, *H.E.S.S.*, which were, and some of them are still, very important for the study of GRB properties. Then, I will review the main results obtained thus far and the still open problems and prospects of GRB astronomy.

**Keywords:** gamma-ray sources; gamma-ray bursts; gamma-ray bursts history; gamma-ray bursts discovery; gamma-ray bursts afterglow discovery; gamma-ray bursts progenitors; gamma-ray bursts cosmology; supernova connection; emission mechanisms; short gamma-ray bursts; neutron star mergers; multissenger astronomy

## 1. Introduction

Gamma ray bursts (GRBs) are among the most intriguing phenomena in the Universe. They are sudden bright flashes of celestial gamma-ray radiation, with variable duration from milliseconds to several hundreds of seconds. Sometimes, their duration achieves thousands of seconds. Most of their emission extends from a few keV up to tens of MeV, but also GeV emission has been detected in many of them, and, in some cases, also TeV emission. With the current X-ray/gamma-ray detectors, their occurrence rate is 2–3 per day over the entire sky. Their arrival times are unpredictable as is their arrival direction. When they occur, their brightness outshines any other celestial X-ray/gamma-ray source.

In this paper I will review the main steps of the research studies on GRBs. In particular, in Section 2 I will start with the GRB discovery with the *Vela* satellites. In Sections 3 and 4 I will discuss the main efforts performed soon after the GRB discovery, mainly with *Venera* satellites, the BATSE experiment and its main results. In Section 5 I will discuss the peculiar story of the *BeppoSAX* discovery of the GRB afterglow and the determination of their extragalactic origin. In Section 6 I will review the immediate consequences of the *BeppoSAX* discovery on the scientific community and on GRB theoretical models, while in Section 7 I will summarize the main other results obtained with *BeppoSAX* on GRBs. Section 8 is devoted to discussing the main results obtained in the post-*BeppoSAX* era. Theoretical aspects, like GRB progenitors and the inner engine, are discussed in Sections 9 and 10. The key role played by GRBs for the birth of the multi-messenger era is reviewed in Section 11, while, in Section 12, the still open issues about GRBs and the importance of GRBs for cosmology and fundamental physics are summarized. In the last

two Sections 13 and 14, I will discuss future prospects for GRB astronomy, particularly the space and ground facilities that are expected to solve many of the still open issues of this new and very intriguing research field.

## 2. GRB Discovery

The earliest GRBs were discovered by chance in 1967 with the American *Vela* satellites. These satellites were a series of 12 military spacecraft with a life time of the order of 1 year, launched between 17 October 1963 (*Vela 1A* and *1B*) and 8 April 1970 (*Vela 6A* and *6B*). The goal of these satellites was to monitor the explosion of nuclear bombs in the terrestrial atmosphere, which were vetoed by the "Partial Test Ban Treaty" issued on 10 October 1963. In particular, with these satellites, the territory of the Soviet Union was monitored, which was the main nuclear-capable state that could perform these tests in the atmosphere. The satellites were spinning and had on-board gamma-ray detectors. The most sophisticated instrumentation was aboard *Vela 5* and *Vela 6*. It included a gamma-ray experiment which consisted of six CsI scintillation crystals, with a total volume of 60 cm$^3$, distributed so to achieve nearly isotropic sensitivity. The energy passband was 0.2–1 MeV in the case of *Vela 5* and 0.3–1.5 MeV in the case of *Vela 6* [1]. The scintillators were shielded for charged particles, thanks to a shield made of high atomic number materials. The time resolution of the data collected from these satellites was continuously improving, from 32 s integration time for *Vela 1* to 64 ms for *Vela 5* and *Vela 6*.

On 2 July 1967, the *Vela 3* and *Vela 4* satellites detected a flash of gamma-ray emission unlike any known nuclear weapon signature. Other similar events were observed with other *Vela* satellites launched later and with better instruments. By analyzing the different arrival times of the bursts as detected by different satellites, the Los Alamos team led by Ray Klebesadel was able to determine rough estimates for the sky positions of sixteen bursts and to definitively rule out a terrestrial or solar origin. The discovery was eventually published in 1973 [1] with the title "Observations of Gamma—Ray Bursts of Cosmic Origin". In Figure 1, one of the GRBs observed with the *Vela* satellites is shown. In the time period of 1969 to 1979, the *Vela* spacecraft (5 and 6) recorded 73 gamma-ray bursts. A preliminary catalog of events was reported by [2].

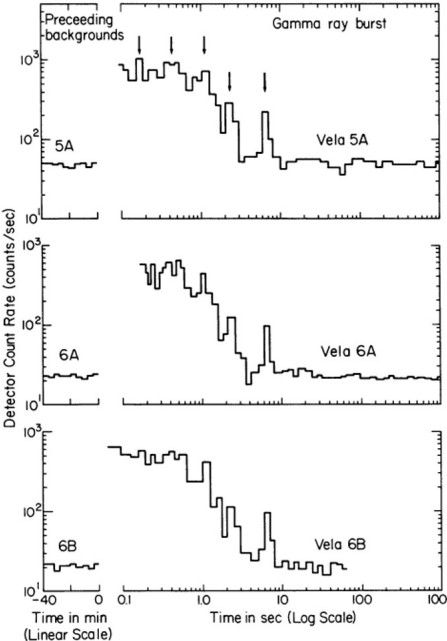

**Figure 1.** One of the 73 gamma-ray bursts observed with the *Vela* satellites. It shows the count rate as a function of time for the gamma-ray burst of 22 August 1970. Arrows indicate some of the common structures. Background count rates immediately preceding the burst are also shown. Reprinted from [1] © AAS. Reproduced with permission.

The first questions were as follows: Which are the sites of these events? Which are their progenitors? Which is the power released in these events? The solution to these issues implied the solution of complex observational problems, like an accurate localization of the events in order to associate them to a possible known source. This was a tough task in the gamma-ray energy band, where the source position could not be accurately determined. A possible solution was the search for GRB counterparts at longer wavelengths.

### 3. Main Efforts Soon after the GRB Discovery

Many satellite missions in the 1970s and 1980s (e.g., the Russian satellites *Venera* 11, 12, 13, 14, Prognoz 6, Prognoz 9, Konus, Granat, and the American Pioneer–Venus Orbiter and Solar Maximum Mission) were launched with instruments also devoted to detect GRBs (see [3,4]). Relevant results about their origin were obtained with the Venera satellites. Aboard *Venera* 11 and *Venera* 12, there was a hard X-ray/soft gamma-ray GRB experiment, *Konus*, developed by the Ioffe Physico–Technical Institute in St. Petersburg, which consisted of six NaI(Tl) scintillator detectors, which were completely open, apart from a shield on the sides and bottom. The detectors were oriented along six different directions and covered all the sky. The different orientation of the detector axes allowed to obtain a localization of the GRB sources with an accuracy $\geq 4$ deg, while, when the mutual distance of the two satellites was also taken into account, a localization of the GRB direction in the arcminute range was even possible through triangulation. In addition to the localization, it was also possible to obtain the temporal structure and photon spectrum of the events [5,6]. A modified version of the *Konus* experiment was also flown aboard *Venera* 13 and *Venera* 14 launched in 1981 October 30 and November 4, respectively. The main differences concerned the number of energy channels and a better time resolution. For example, the temporal accumulation of the photon spectra was 0.5 s instead of 4 s.

The *Konus* results on GRB were very interesting. A time-resolved correlation was found between the source intensity and peak energy (interpreted as bremsstrahlung temperature) of the $EF(E)$ spectrum, where $E$ is the photon energy and $F(E)$ is the GRB energy spectrum. In addition, the earliest evidence was found of an isotropic distribution of the GRB positions in the sky [6,7] as shown in Figure 2. This distribution could mean that the GRBs are either very close to the Earth or very far.

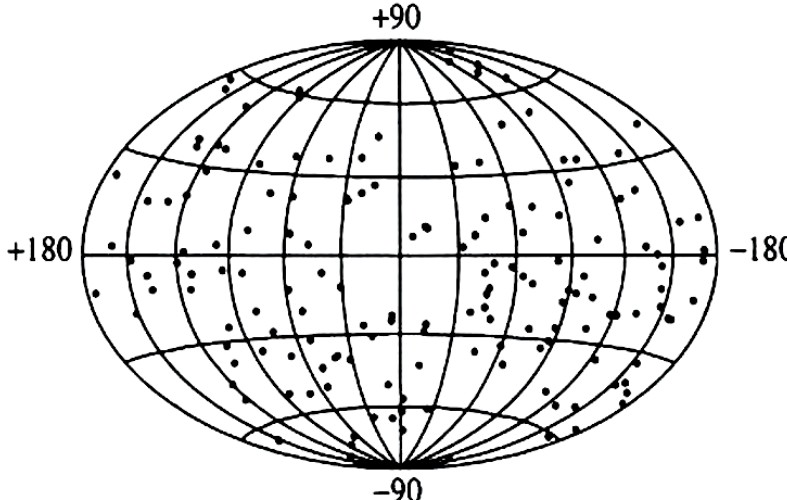

**Figure 2.** Distribution of GRBs in the sky in Galactic coordinates as obtained with the Konus experiment aboard the *Venera* 11–14 missions. Reprinted from [7].

### 4. The BATSE Era

The definitive answer to the GRB distribution in the sky was given by the BATSE (Burst And Transient Source Experiment) experiment aboard the American satellite *Compton Gamma Ray Observatory*. Their isotropy in the sky was definitely confirmed (Figure 3).

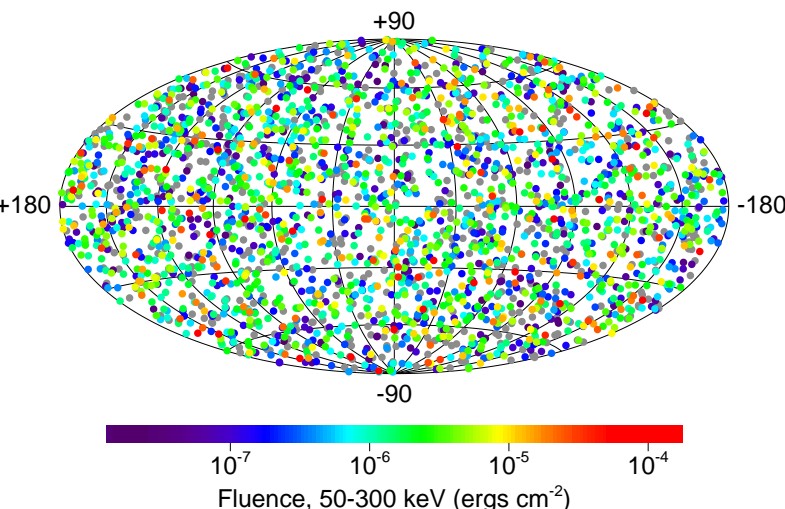

**Figure 3.** Sky distribution in galactic coordinates of 2704 GRBs detected with the BATSE experiment aboard the *Compton Gamma Ray Observatory* satellite. Adapted from [8] © AAS. Reproduced with permission. (Credits: https://gammaray.nsstc.nasa.gov/batse/grb/skymap/ 3 May 2024).

The *Compton Gamma Ray Observatory* mission (e.g., [9]), designed in the 1980s, was launched on 5 April 1991 by the Space Shuttle Atlantis and reentered the Earth atmosphere on 4 June 2000. Three hard X-ray/soft gamma-ray experiments were on board: an Oriented Scintillation Spectrometer Experiment (OSSE), a COMPton TELescope (COMPTEL), and BATSE. This experiment, whose PI was Gerald J. Fishman, a scientist of the NASA Marshall Space Center in Huntsville (see [10]), was developed mainly to detect and locate GRBs along with the study of their temporal and spectral properties. It consisted of eight completely open NaI(Tl) Large Area Detectors (LADs) at the corners of the spacecraft, each sensitive in the 30 keV–2 MeV energy range and with an exposed area of 2025 cm$^2$. For each LAD, there was a smaller spectroscopy detector (SD) with a detection area of about 600 cm$^2$ [11], optimized for energy resolution and broad energy coverage (10 keV–11 MeV). The GRB shape could be transmitted with different time resolutions, down to μs time scales. BATSE was capable of establishing not only the isotropic distribution of GRBs in the sky, but also many other properties of the GRB prompt emission, like the bimodality of the GRB prompt emission duration $T_{90}$ (see Figure 4), the spectral distribution of the prompt emission (see Figure 5) described by a smoothed broken power law (also called the Band function from the paper by Band et al. [12], who proposed it), and the GRB intensity distribution (see Figure 6).

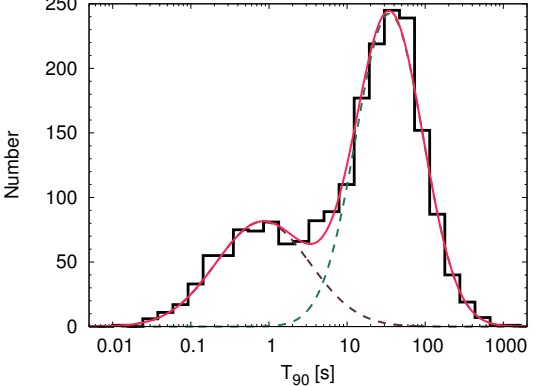

**Figure 4.** Distribution of $T_{90}$ for 2006 GRBs detected with BATSE with significant $T_{90}$ estimates. $T_{90}$ is defined as the time during which the GRB cumulative counts increase from 5% to 95% of the total detected counts. Dashed lines give the best fit to the $T_{90}$ distribution.

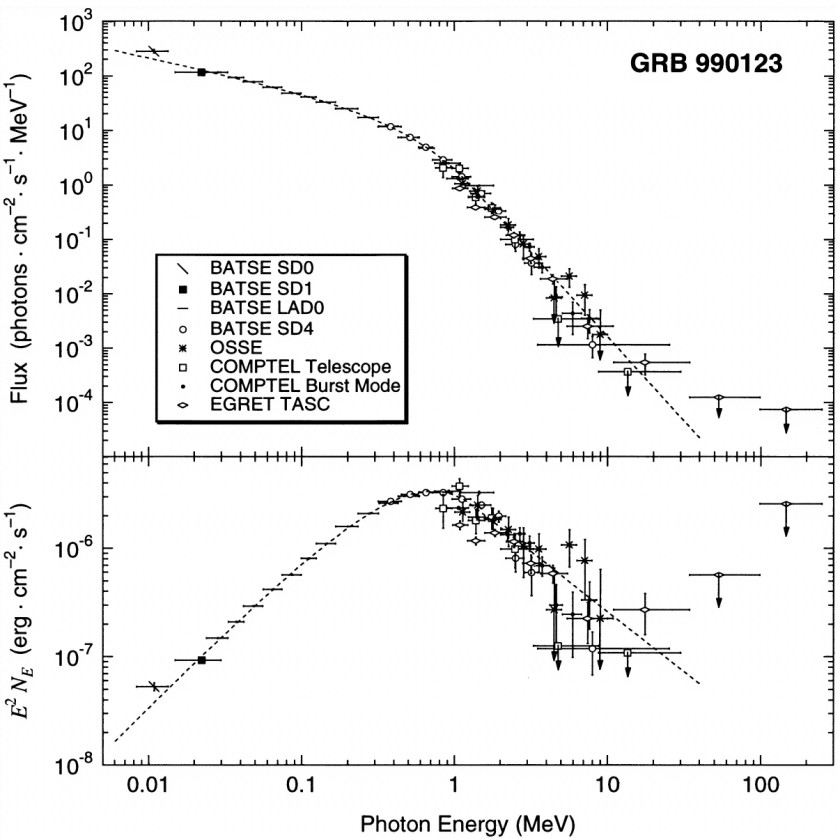

**Figure 5.** An example of GRB photon spectrum and the corresponding $E^2N(E)$ function. It shows the BATSE spectrum of the GRB occurred on 1999 January 23 (GRB 990123), that was also detected and promptly accurately localized with *BeppoSAX*. Reprinted from [13] © AAS. Reproduced with permission.

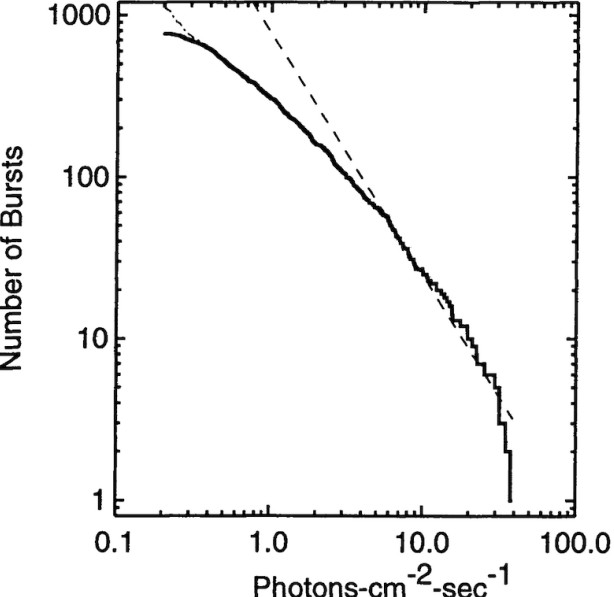

**Figure 6.** Integral log $N$–log $P$ distribution of 772 GRB detected with BATSE in the range 50–300 keV band. The peak flux $P$ is integrated over 1.024 s. The dot-dashed line gives the correction for instrumental trigger efficiency, while the dashed line gives the power-law slope ($-3/2$) expected in the case of an homogeneous distribution of GRBs in space. Reprinted from [14].

In spite of these and many other results, due to the uncertainty in the BATSE GRB localization capability (several degrees), all the numerous efforts to discover X-ray or optical

or radio counterparts of the BATSE GRBs were unsuccessful, and the sites of the GRBs were not discovered.

The uncertainty in the GRB sites, and thus in their distance scale, was also a matter of debate as demonstrated by the meeting that took place at the Baird Auditorium of the Smithsonian museum of Natural History in Washington DC (USA) on 22 April 1995, where there were two main points of view about the GRB sites and progenitors, one led by Bodhan Paczynski and the other led by Don Lamb. The viewpoint of Paczynski was the following [15]: "At this time, the cosmological distance scale is strongly favored over the Galactic one, but is not proven. A definite proof (or dis-proof) could be provided with the results of a search for very weak bursts in the Andromeda galaxy (M31) with an instrument 10 times more sensitive than BATSE. If the bursters are indeed at cosmological distances then they are the most luminous sources of electromagnetic radiation known in the Universe. At this time we have no clue as to their nature, even though well over a hundred suggestions were published in the scientific journals".

Instead, Don Lamb had a completely different point of view [16]: "We do not know the distance scale to gamma-ray bursts. Here I discuss several observational results and theoretical calculations which provide evidence about the distance scale. First, I describe the recent discovery that many neutron stars have high enough velocities to escape from the Milky Way. These high velocity neutron stars form a distant, previously unknown Galactic corona. This distant corona is isotropic when viewed from Earth, and consequently, the population of neutron stars in it can easily explain the angular and brightness distributions of the BATSE bursts".

Thus, in 1995, in spite of some convictions, the distance scale of the GRB sites was still an open issue, with the following premonitory conclusion of the debate by Martin Rees [17]: "I'm enough an optimist to believe that it will only be a few years before we know where (and perhaps even what) the gamma-ray bursts are".

## 5. The *BeppoSAX* Afterglow Discovery

The solution of the mystery about the GRB sites was obtained with the Italian *BeppoSAX* satellite with Dutch participation [18]: in seven months from the beginning of its operational phase (October 1996), thanks to *BeppoSAX*, it was possible to establish that GRBs are huge explosive events in galaxies at cosmological distances. Why did *BeppoSAX* solve the mystery of the GRB distance? This story merits description, recalling the evolution of the *BeppoSAX* science goals and its design.

### 5.1. SAX *Initial Goals and Evolution in Its Configuration*

Actually, the initial main goals of this satellite, initially named SAX (Satellite Astronomia X, in Italian), had as its initial main goals the following: (1) the study of celestial X-ray sources in a broad energy band (0.1–300 keV) with narrow field instruments; (2) the X-ray (2–30 keV) monitoring of the sky, in particular, of the galactic plane, with wide field cameras.

The reason for such science goals for SAX was that, when SAX was proposed and approved in 1981 by the National Space Plan (later, in 1988, Italian Space Agency), X-ray astronomy was 20-year aged, with the best knowledge of the X-ray sky mainly obtained with satellite missions in the low energy band (<10 keV). The maximum sensitivity and localization were achieved in the softer X-ray band (<3 keV) with the introduction of grazing incidence optics in satellite missions, like *Einstein* [19]. The celestial hard X-ray band (>20 keV) was explored mainly with balloon-borne experiments, and source spectra were well measured only for the strongest sources. With the growing in the sensitivity, the source spectra appeared more complex, and a source variability was detected at all time scales and at all energies. But a systematic study of source time variability, along with the covering of a broad energy band with high sensitivity, was still not performed. Thus, the main target of SAX was that of covering, with narrow field instruments, an unprecedented energy range with a sensitivity as balanced as possible, and, at the same time, monitoring

wide portions of the sky for variability studies. GRBs were only mentioned as possible X-ray targets for wide field cameras (see below) for the possible affinity of GRBs with emission from neutron star binaries, on the basis of the discovery, with the *Venera* 11 satellite, of the famous flaring source in Dorado firstly observed on 5 March 1979 [20] but later recognized as a different class of sources (Soft Gamma-Ray Repeaters or magnetars for their very strong magnetic field).

To achieve the mentioned goals, the initial SAX payload proposed included three instruments, two with a narrow field of view (FOV) and one with a wide FOV. The narrow field instruments (NFIs), oriented in the same direction, were as follows:

- A Gas Scintillator Proportional Counter (GSPC) with 2–35 keV energy passband, surmounted by a coded mask (3 deg FOV) with arcminute imaging capability. The Principal Investigator (PI) was Giuseppe Manzo from the CNR (Italian National Research Council) Institute of Cosmic Physics and Informatics.

- A Phoswich Detection System (PDS) at higher energies (15–300 keV), consisting of four independent detection units, each one made of a sandwich of NaI(Tl) plus CsI(Na) scintillator crystals. The NaI(Tl) was used as the main detector, with the CsI(Na) as an active shield from the bottom. This technique, called phoswich (=PHOSphor sand-WICH), had been demonstrated to provide a very low instrument background (BKG). To further reduce the BKG, four slabs of CsI(Na) detectors, in anti-coincidence with the four phoswich units, laterally covered the instrument. The PDS FOV, of 1.5 deg (Full Width at Half Maximum, FWHM), was obtained by means of honeycomb collimators. The PI was myself, at that time a scientist of the CNR Institute of Technology and Study of the Extraterrestrial Radiations. See Frontera et al. [21].

The wide field instrument consisted of two wide field cameras (WFCs), with axes perpendicular to the NFI and in opposite directions to each other. Each camera was made of a proportional counter surmounted by a coded mask. The energy band of each WFC was 2–28 keV, and its FOV was $20° \times 20°$ (FWHM) with imaging capability with an angular resolution of 3–4 arcmin. The PI was Rick Jager from the Institute of Space Resarch (SRON) in Utrecht (Holland). See Jager et al. [22].

The rationale to have on board the same mission two WFCs, oriented along two opposite directions perpendicular to NFIs, was to have the widest FOV for the monitoring of a large number of variable sources, in particular, new transients, and to have the possibility to perform shorter pointed observations at particular states of them.

During the *industrial phase A* study performed by the AERITALIA (later, Alenia Spazio), the GSPC was replaced by a set of four focusing telescopes having the same on-axis direction, with one of them (Low Energy Concentrator Spectrometer, LECS) with a passband from 0.1 to 10 keV, and the other three (Medium Energy Concentrators Spectrometers, MECS) with a passband from 1.3 to 10 keV. All of them had a FOV of 0.5 deg and an angular resolution of the order of 1 arcmin. The PI of LECS and MECS was Giuliano Boella from the CNR Institute of Cosmic Physics, with responsibility of the LECS focal plane detector by Arvind Parmar from ESA-ESTEC, Noordwijk (The Netherlands). See Boella et al. [23], and Parmar et al. [24].

Given the high scientific interest at that time for the cyclotron line features in the spectra of X-ray pulsars, the GSPC instrument, for its very good energy resolution, was still included in the payload with no coded mask and a higher (5 atm) gas pressure (HPGSPC). Its final FOV was 1.1° FWHM, and its energy passband from 4 to 60 keV. See Manzo et al. [25].

The original proposal did not include GRBs as a main science goal. This was due to the fact that in the early 1980s, our knowledge of GRBs was strictly confined to gamma-rays or very hard X-rays. It was not clear whether an X-ray phenomenology could be expected. For sure, until that time, no transient X-ray detected phenomenon had been associated with a GRB. Moreover, due to the wide spread of interpreting models, it was not clear whether any X-ray delayed emission could be detected.

### 5.2. Addition of a Gamma-Ray Burst Monitor GRBM and Establishment of a Team for GRBs Identification and Localization

In 1984, during the phase A study mentioned above, I proposed to use the four slabs of CsI(Na) anti-coincidence detectors as a Gamma-Ray Burst Monitor (GRBM) with a passband from 40 to 700 keV (see Internal Report of CNR-TESRE, No. 99, 1984). The motivation for this implementation was that the axis of two of these shields was parallel to that of the two WFCs. This feature was very suggestive. Indeed, we expected that about three GRBs per year (see SAX Observers' Handbook, Issue 1.0, 1995) could enter into the common field of view of WFCs and GRBM, and thus they could be identified as true GRBs by GRBM and localized by WFCs within 3–5 arcmin, an accuracy never achieved at that time by the GRB monitors already launched or only designed. The GRB identification with GRBM was crucial because transient events detected with only WFCs could be originated by other phenomena (e.g., star flaring and X-ray bursts), while their contemporary detection in the GRBM energy band (>40 keV) was an almost certain signature of being true GRBs. Obviously, it was required to develop a proper electronic chain, an in-flight trigger system, and a prompt download of the data. The GRBM proposal had also an international resonance [26].

In 1990, a cheaper version of the GRBM proposal was approved by the Italian Space Agency (ASI) as a further instrument of SAX. With this addition, the final SAX payload became that shown in Figure 7. Once the GRBM proposal was approved, its design was better defined and improved, in the context of limited resources, and without interfering with the anti-coincidence function of the CsI(Na) slabs, which had been the primary driver of the PDS detector design. For a full GRBM description, see Ref. [27].

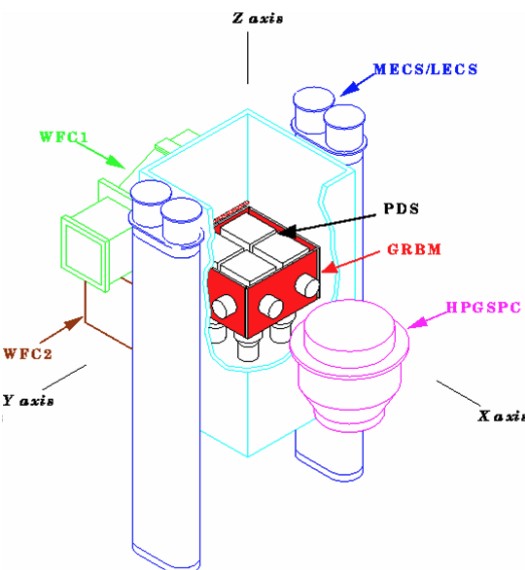

**Figure 7.** The final *BeppoSAX* payload. **GRBM** is shown in red as part of the PDS instrument. It is made of the four independent slabs of CsI(Na) scintillators, 1 cm thick. They were initially foreseen as active anticoincidence shields of the 4 phoswich units. Notice that two opposite GRBM units were oriented as the two WFCs.

By means of a Monte Carlo code, we found that the different orientations of the GRBM units could be exploited for obtaining a crude GRB localization that was sufficient for deconvolving the GRB count spectra [28]. The implementation of an optimized response function of GRBM required a very detailed description of the entire SAX satellite, which was obtained with both simulations and ground calibrations. The simulations were performed thanks to the development of a very detailed Monte Carlo code [29], while the ground calibrations were performed at different steps, with the last one having a calibration campaign performed at ESTEC (Noordwijk, The Netherlands) after the integration of the instrument in the satellite [30].

Before the SAX launch, in response to an international call by the SAX collaboration, the established PDS/GRBM team submitted a proposal to obtain WFCs data in the case of GRBs identified with GRBM. The goal was to promptly and accurately localize GRBs and perform a follow-up of the localized event with SAX Narrow Field Instruments (NFIs). Also, the BATSE team submitted a similar proposal for GRBs identified with the BATSE experiment. Both proposals were approved by the SAX Time Allocation Committee.

### 5.3. The SAX Launch and the First Detected GRBs

SAX was launched on 30 April 1996 from Cape Canaveral with an Atlas-Centaure double rocket, reaching an altitude of 600 km with a final orbit inclination of 3.8 deg (see Figure 8). Soon after the successful launch, the satellite name was modified, becoming *BeppoSAX* in honor of the famous Italian physicist Giuseppe (called Beppo by his friends) Occhialini (see https://en.wikipedia.org/wiki/Giuseppe_Occhialini (accessed on 2 May 2024)).

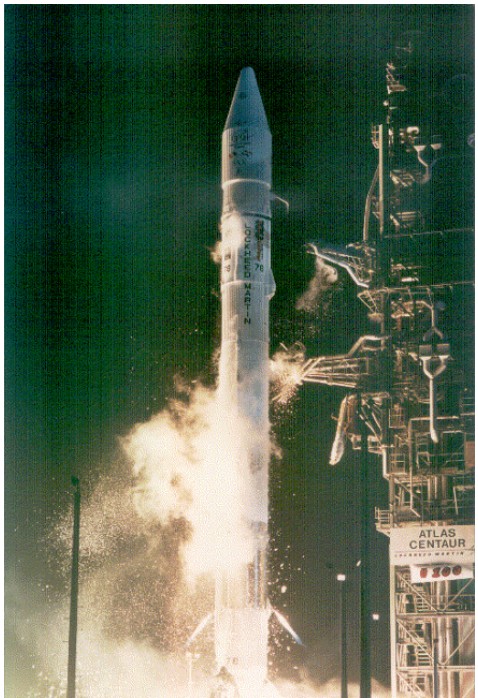
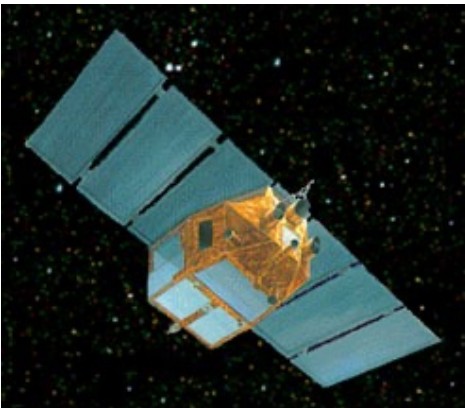

**Figure 8.** (**Left**): SAX in the Cape Canaveral launching pad before its launch with an Atlas-Centaure double rocket. (**Right**): an artistic view of the satellite in flight.

The commissioning phase had a two month duration up to the end of June 1996, while the Science Verification Phase (SVP) had a 3-month duration (July–September 1996). The satellite started its operational phase in October 1996. The telemetry link (10 min per orbit) with the satellite was obtained by means of the ASI ground station in Malindi, which allowed to download the mass memory with the satellite data and to up-link telecommands.

The first GRB that we identified with GRBM and found in the field of view of one of two WFCs, occurred during the SVP phase, on 20 July 1996. However, it could be accurately localized only 20 days after the event time, and, thus, after about one month, the *BeppoSAX* NFIs were pointed to the GRB direction, with no detection of an X-ray counterpart [31].

From this result, it was clear that a possible residual X-ray radiation, if any, could have been found only in the case that it was possible to point, as soon as possible, the NFIs along the direction of well-localized events. To this end, we optimized all the procedures to minimize the time needed to establish whether a GRBM event was a true GRB, whether a contemporary source image was detected by WFCs, and, in the positive, to submit the request of a prompt *BeppoSAX* Target of Opportunity (TOO) to re-point the NFIs toward the localized event.

The first GRB detected and localized with this procedure occurred on 11 January 1997. The light curve of this event (GRB 970111) is shown in the top panel of Figure 9. Its earliest position was derived with a 10 arcmin error radius (see bottom panel of Figure 9). On the basis of this result, an X-ray follow-up with NFIs was started 16 hrs later. The same derived position was given to our collaborators for the search of an optical/radio counterpart. In the WFC error box, Dale Frail and his colleagues, with the VLA (Very Large Array) radio telescope in Socorro (NM, USA), observed an unusual radio source (VLA1528.7 + 1945), variable on time scales of days. A similar variability was never found before in a radio source. Thus, the source was assumed to be the likely radio-counterpart of the GRB event, and a paper was promptly submitted to the *Nature* journal. However, after about 20 days, with a better deconvolution function, we obtained a more accurate WFC position and error box. The error radius became only 3 arcmin, and the centroid position was about 4 arcmin far from the previous one (see bottom panel of Figure 9). The radio source was no more consistent with the new WFC position. Similarly, the two bright X-ray sources found with the *BeppoSAX* NFIs [32], with one of them coincident with the radio source position, were not compatible with the new WFC position. The paper submitted to *Nature* was withdrawn and later published in *ApJ* [33].

In the current time of multi-messenger astromony (see later), the experience acquired with GRB 970111 is an important teaching. A very accurate positional coincidence is crucial for associating a GRB event to an optical/IR counterpart or to a gravitational wave (GW) signal or to a neutrino event, and viceversa. A mission concept like *THESEUS* (see later), thanks to its accurate capability to localize a GRB in X-rays and to perform a prompt follow-up with the infrared telescope onboard, appears to be the best solution for the prompt association of a GRB event to an optical counterpart. Also, the association of a gravitational wave (GW) signal or a neutrino event to a GRB should follow similar strategies.

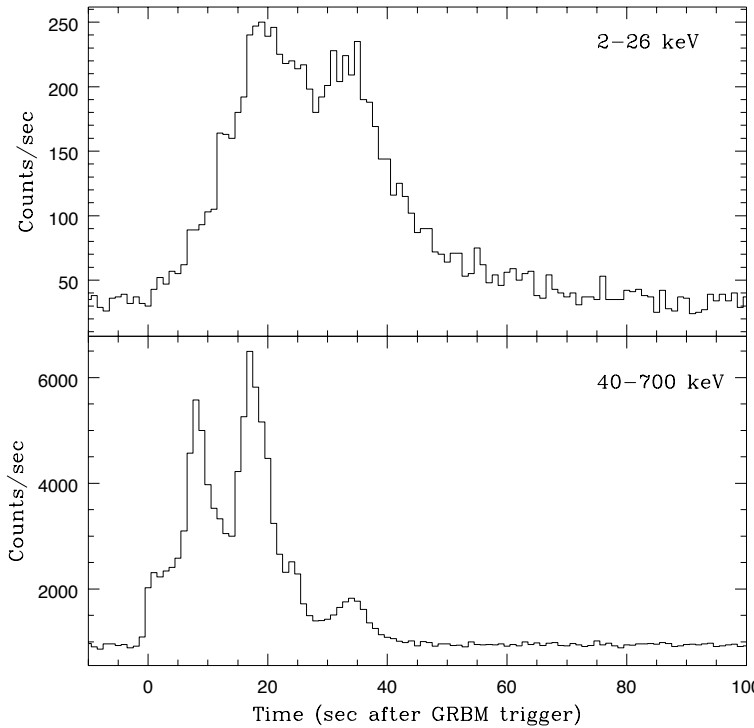

**Figure 9.** *Cont.*

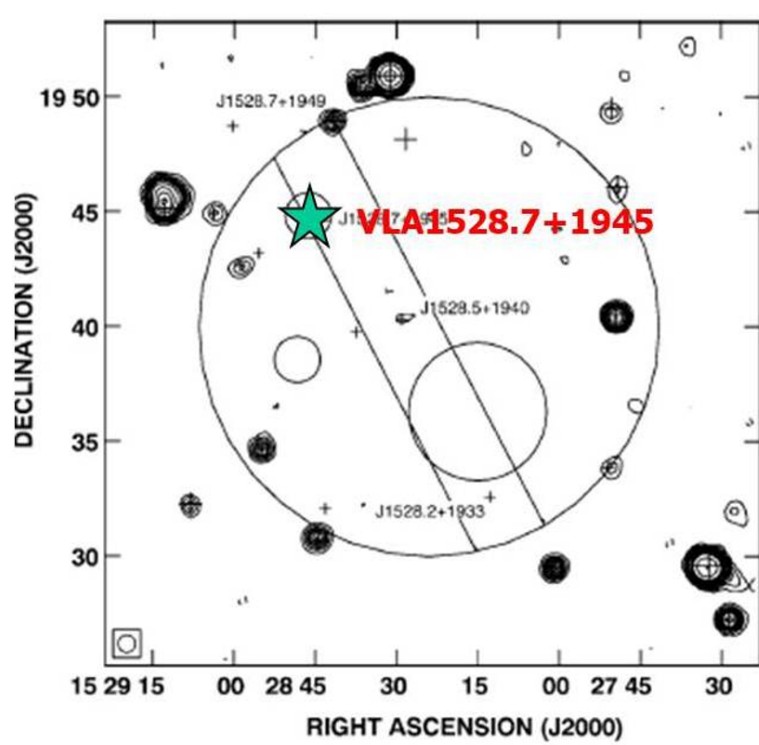

**Figure 9.** (**Top panel**): Light curve of GRB 970111. Reprinted from [32]. (**Bottom panel**): The largest circle gives the earliest WFC error box, while the smaller circle gives the latest WFC error box. The strip crossing the largest circle is the error box derived with the Interplanetary Network (IPN). As can be seen, in the intersection of the refined WFC error box with the IPN annulus, no source was observed. Adapted from the figure of [33].

*5.4. The First GRB Afterglow Discovery*

The mystery about the GRB sites was solved with the discovery of the X-ray and optical counterpart of GRB 970228. This GRB, identified with GRBM and localized with one of the two WFCs, showed a light curve with a bright peak, followed by a train of three more peaks of decreasing intensity (see top panel of Figure 10).

The NFI follow-up was performed 8 h after the time of the event and lasted about 9 h. In the MECS FOV, a previously unknown source (SAX J0501.7 + 1146) was found with a flux of $(2.8 \pm 0.4) \times 10^{-12}$ erg cm$^{-2}$s$^{-1}$ in the 2–10 keV band. The source was pointed out again three days after, and it was found to have faded by a factor 20 (see bottom panel of Figure 10).

By subdividing the first MECS observation into three subsets, the time behavior of the discovered source was derived. It resulted that the source was fading according to a power law $N(t) \propto t^{-1.33}$, where $t$ is the time since trigger. Also, the flux detected by the WFCs during the GRB tail of the light curve, in the same energy band of MECS, was found to be consistent with the fading law measured with NFIs. This fact is crucial to state that the X-ray source found was the delayed radiation (i.e., the **afterglow**) of the GRB event [34].

In parallel, first the GRB coordinates obtained with WFC and later those obtained with NFIs, were given by the GRBM team to a few European Observatories working in service, and, in parallel, distributed through IAU circulars (see [35]). Various observers performed optical observations.

Promptly, three optical observations of the distributed position were performed by the Dutch group led by Jan Van Paradijs. With the same filter, the first observation was performed with the William Herschel Telescope on 28 February, and the other two observations were performed on 8 March, one with the William Herschel and the other with the

Isaac Newton Telescope. All of these observations showed the presence of a previously unknown optical source that was fading (see Figure 11) from $V = 21.3$ to $V > 23.6$ [36].

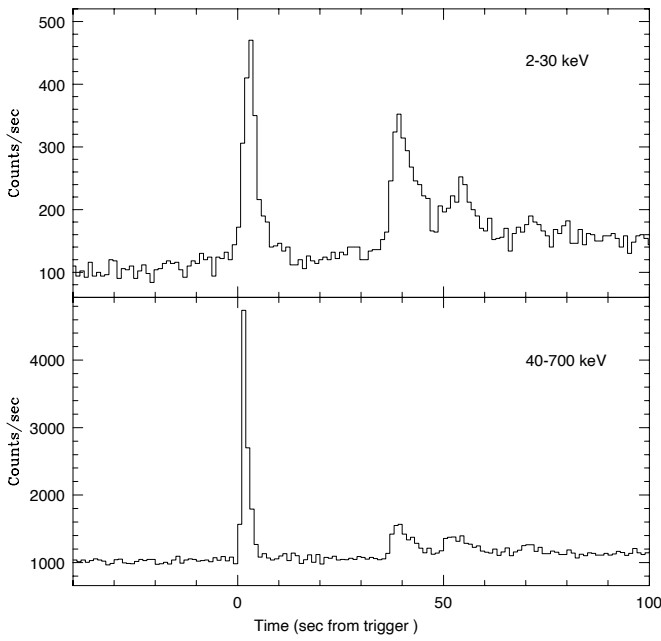

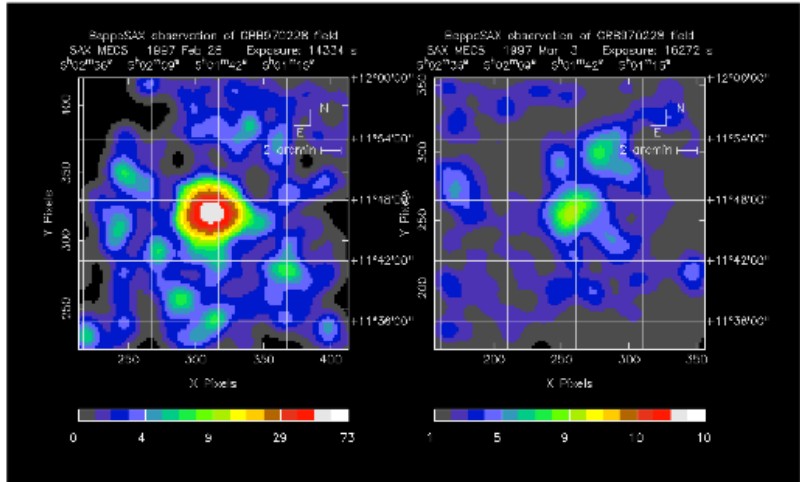

**Figure 10.** (**Top panel**): Light curve of GRB 970228 as detected with WFC and **GRBM**. (**Bottom panel**): X-ray images of GRB 970228 as detected with MECS telescopes at two different times. The image on the left is that obtained 8 to 16 h after the burst. The image on the right is that obtained in the observation 3.5 days after the burst. From the first to the second X-ray observation, the source had faded by a factor 20. Reprinted from [34].

Given the error box (~50 arcsec radius) associated with SAX J0501.7 + 1146, to further confirm the association of this source with the GRB event, we proposed an observation with the High-Resolution Imager (HRI) aboard the X-ray satellite *ROSAT*. This instrument, thanks to its sensitivity and high angular resolution (10 arcsec radius) in the energy band 0.1–2.4 keV, could provide a better position and a much lower error box of the source direction. The observation was performed on March 10 and lasted three days. Eight sources were detected in the HRI FOV (20 arcmin) [37], with only one (RXJ050146 + 1146.9) in the error box of the X-ray source found with the *BeppoSAX* NFIs. The source position (see right panel of Figure 12) was coincident, within 2 arcsec, with the discovered optical transient [36]. In addition its intensity was found to be fully consistent (see left panel of

Figure 12) with the extrapolation, at the *ROSAT* observing time, of the power law decay estimated from the *BeppoSAX* LECS afterglow spectrum in the 0.1–2.4 keV energy band. With these results, the association of the *ROSAT* and *BeppoSAX* sources with the afterglow of GRB 970228 became conclusive.

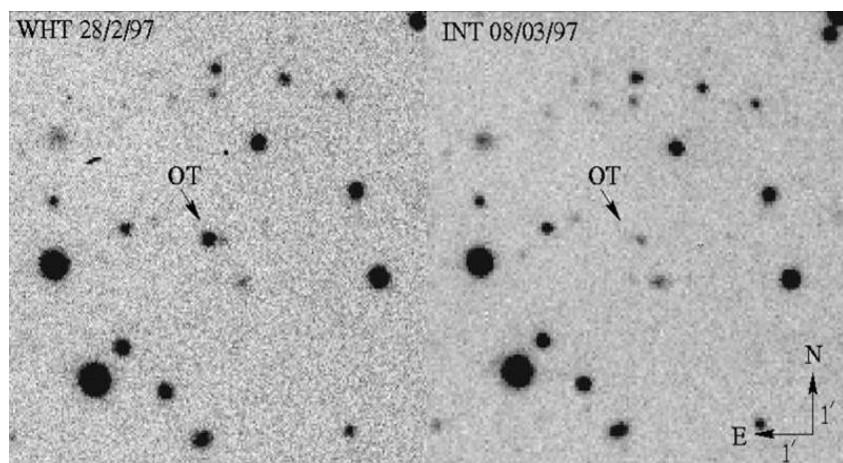

**Figure 11.** Discovery of the optical counterpart of GRB 970228 at two different times (28 February 1997, and 8 March 1997). The source is clearly fading. Reprinted from [36].

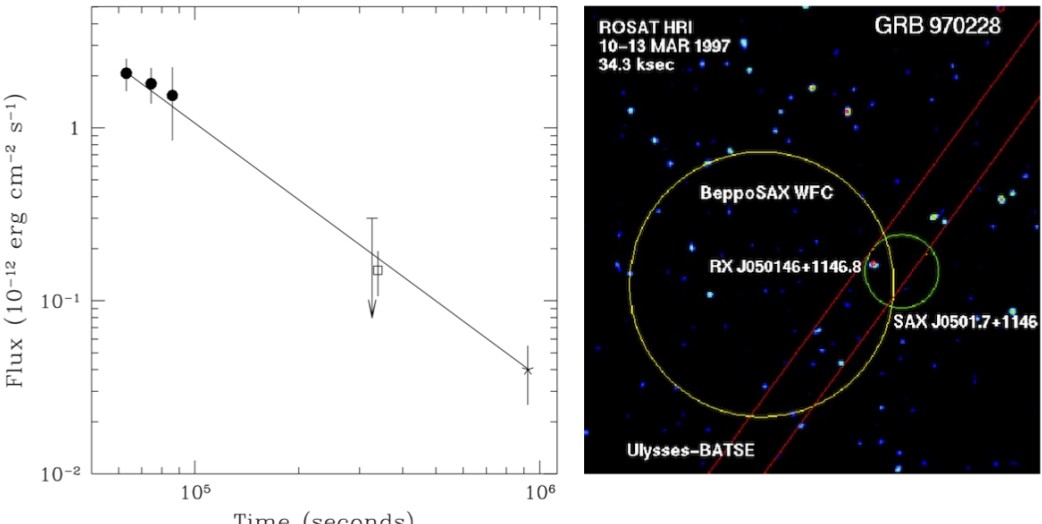

**Figure 12.** (**Left**): Decline in the GRB 970228 flux in 0.1–2.4 keV with time, starting from the follow-up time of GRB 970228 with *BeppoSAX*, uncorrected for galactic absorption. The filled dots are the LECS data points, the arrow is the LECS $3\sigma$ upper limit, the square gives the flux extrapolated from the MECS detection, and the asterisk shows the *ROSAT* HRI data point. The best fit power-law decay (continuous line) is also shown. (**Right**): Image of the *ROSAT* HRI FOV (8 arcmin wide) found during the GRB 970228 follow-up performed on 1997 March 10. Only one source (RXJ050146 + 1146.8) was found inside the error box (small circle with $\sim$ 50 arcsec radius) of the fading source SAX J0501.7 + 1146. In addition, this *ROSAT* source was coincident with the optical source (in red) within 2 arcsec. The large circle shows the 3 arcmin error circle of GRB 970228 as determined with the *BeppoSAX* WFC, while the two straight lines give the uncertainty strip derived from the *BeppoSAX* GRBM and *Ulysses* timings of GRB 970228. Reprinted from Frontera et al. [37].

A detailed spectral analysis of GRB 970228 and its afterglow was also performed [38]. While the spectrum of the prompt emission was consistent with the Band function and showed, within each peak, a hard-to-soft evolution, the afterglow spectrum was a stable power law ($N(E) \propto E^{-2.04}$).

Thus, as a result of these X-ray observations, the temporal and spectral properties of the afterglow were in favor of a non-thermal process (e.g., [39]) and became the basic building block for GRB theories.

Thirty-nine days after the burst, a further observation of the GRB 970228 optical counterpart was performed with the *Hubble Space Telescope*. It was found that the point-like source had further faded down to $V = 26.4$. In addition, it appeared to be embedded in a faint nebular source (see Figure 13) with $V \approx 25$ and an extension of $\sim 1$ arcsec, likely, but yet not necessarily, a host galaxy [40].

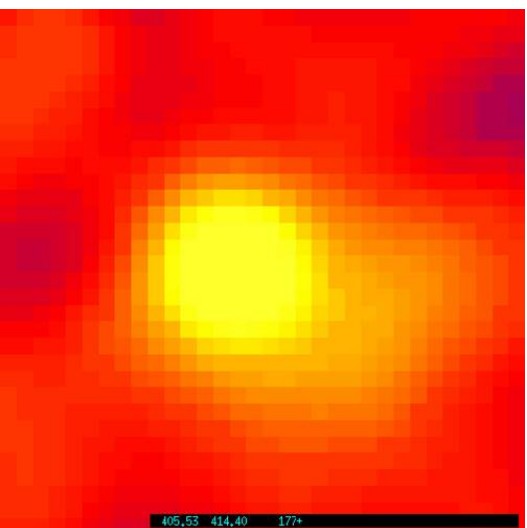

**Figure 13.** Optical image of the GRB counterpart obtained with *Hubble Space Telescope*, in an observation performed 39 days after the burst. The point-like source had further faded down to $V = 26.4$ and it seemed to be embedded in a faint nebular source with an extension of $\sim 1$ arcsec and a magnitude $V \approx 25$, likely, but yet not yet necessarily, a host galaxy. Figure provided by A. Fruchter, see [41].

*5.5. The First Measurement of a GRB Distance*

The further turning point of the *BeppoSAX* discoveries occurred on 8 May 1997, when GRB 970508 was identified with GRBM and localized with one of the two WFCs [42,43]. The light curve of the event as detected with both instruments is shown in the left panel of Figure 14 [44]. About 6 h after the event, the field obtained with WFC was acquired by NFIs, and an unknown X-ray source (1SAX J0653.8+7916) was detected. Its coordinates were promptly distributed [42], an optical source was discovered [45], and soon later, several groups performed optical observations of the discovered source. The detected optical counterpart showed a flux that was increasing after two days, arriving at a magnitude of $R = 20.14$ and then starting to fade with the power law (see [44] and references therein).

On 11 May, when the optical afterglow was still relatively bright, our collaborators of the CalTech/NRAO group observed it with the Keck Low Resolution Imaging Spectrograph. Various absorption lines were identified (see right panel of Figure 14): some at redshift $z = 0.835$, and some others at redshift of $z = 0.767$ [46]. Weeks later, when the point-like object was almost invisible, the highest redshift value ($z = 0.835$) was found in the emission lines from an extended source in the same position of the point-like object: the galaxy that had hosted the fading object. The mystery of the GRB sites was solved. Remote galaxies harbor GRBs.

The immediate consequence of this discovery was that it was eventually possible to fix the energy scale. From the luminosity distance of the GRB 970508 optical counterpart ($1.49 \times 10^{28}$ cm), obtained assuming a standard Friedmann cosmology with $H_0 = 70 \text{ km s}^{-1} \text{ Mpc}^{-1}$ and $\Omega_0 = 0.2$, it was possible to derive the first estimate of the energy released by a GRB: $E_{iso} = (0.61 \pm 0.13) \times 10^{52}$ ergs, assuming an isotropic emission.

GRB 970508 was also relevant for the discovery, with the VLA radio telescope, of the first radio afterglow [47]. The radio emission from the optical GRB counterpart showed a phenomenon known as *scintillation* that derives from the effects of interstellar clouds on sources of a very small angular size. In GRB 970508, the scintillation disappeared after about two months. From the source angular size and its distance, it was possible to derive the expansion velocity of the radio source [47]. It resulted to be around 2*c*, an apparent superluminal expansion, typical of sources expanding at relativistic velocity.

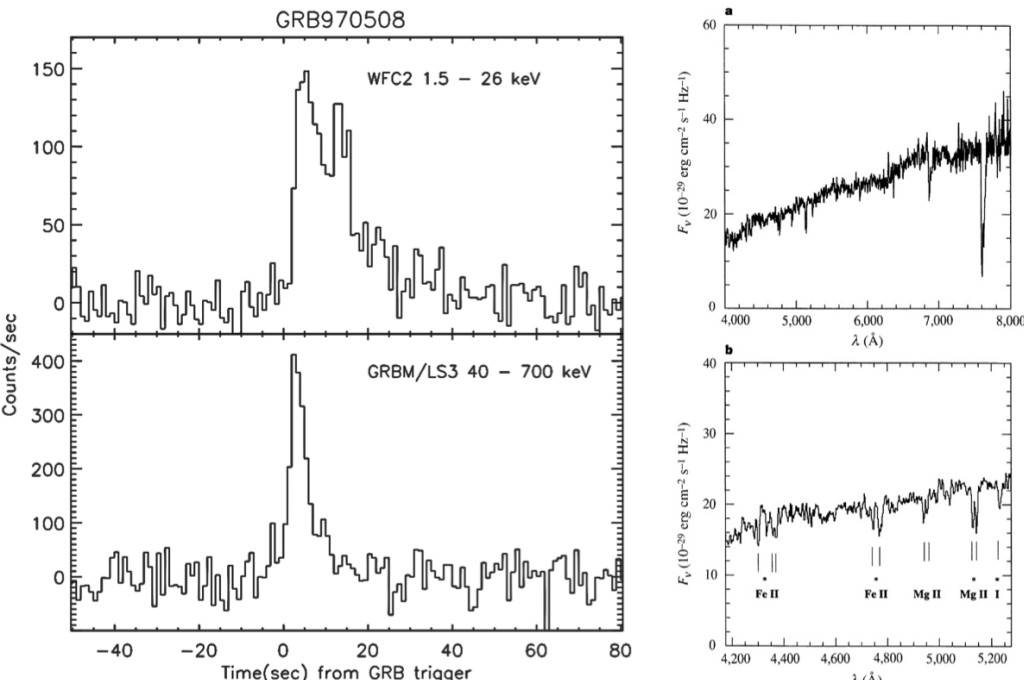

**Figure 14.** (**Left top**): Light curve detected by WFCs (1.5–26 keV). (**Left bottom**): Light curve detected by GRBM (40–700 keV). Reprinted from [44]. (**Right**): Spectrum of the optical counterpart of GRB 970508 taken with the Keck Low Resolution Imaging Spectrograph [46]. (**a**) Full spectrum; (**b**) expansion of a limited region of the spectrum, with strong absorption lines. Various absorption lines were discovered: those marked with an asterisk correspond to a system at redshift $z = 0.835$, while the others at redshift of $z = 0.767$. The largest $z$ values were found, weeks later, in the emission lines from the host galaxy associated with the fading object.

On 14 December 1997, also the redshift of another *BeppoSAX* GRB (971214) was measured: z = 3.42. The corresponding energetics was $(2.45 \pm 0.28) \times 10^{53}$ ergs corresponding to 0.14 $M_\odot c^2$, a value never observed before.

## 6. Immediate Consequences of the *BeppoSAX* Discoveries

### 6.1. Scientific Community Reaction

- In the first two years (1997–1998), the number of papers citing *BeppoSAX* was similar to those citing *Hubble Space Telescope* (about 200/yr) (see Figure 15). The Science/AAAS journal classifed GRB discoveries among the top ten over the world and over all the science fields.
- The data flow from the *INTEGRAL* satellite was modified for a prompt localization of GRBs through an on-ground data analysis software of the IBIS instrument [48].
- NASA issued an Announcement of Opportunity for a new medium-size satellite mission, that led to the *Swift* selection (now *Neil Gehrels Swift Observatory*). The *Swift* mission, still operational, has a configuration similar to that of *BeppoSAX*, with a wide field GRB monitor (BAT (Burst Alert Telescope), [49]) for the prompt identification and localization of GRBs, and an X-ray Telescope (XRT, [50]) plus an Ultraviolet/Optical Telescope (UVOT, [51]) for the afterglow observation. In order to study the early

afterglow, impossible with *BeppoSAX* (anything was known about the GRB evolution), the GRB follow-up is automatically performed in a very short time (∼100 s) [52].

- All scientists who managed large radio and optical telescopes devoted observation time to follow-up GRBs localized with *BeppoSAX*. Also the observation procedures and equipment were changed to make these observations faster.
- Several new optical and/or NIR telescopes were built or modified to allow the robotic pointing of the *BeppoSAX* GRB events.
- The *BeppoSAX* GRB coordinates were distributed through the already existing GCN (General Coordinates Network) circulars set up by NASA, which received an impressive boost by the *BeppoSAX* findings.
- Also, the *Fermi* high-energy gamma-ray satellite was designed taking into account the *BeppoSAX* payload configuration: a Gamma-Ray Burst Monitor *GBM* (8 keV–40 MeV over the full unocculted sky) to identify GRBs, and a *LAT* gamma-ray telescope (20 MeV–300 GeV, 2.3 sr FOV) to localize and study them in the MeV/GeV energy range [53,54].
- A similar design to *BeppoSAX* was adopted for the *AGILE* Italian satellite, with a hard X-ray imager (Super*AGILE*) sensitive in the range 18–60 keV with about 1 sr FOV and a gamma-ray imager sensitive in the range 30 MeV–50 GeV [55].

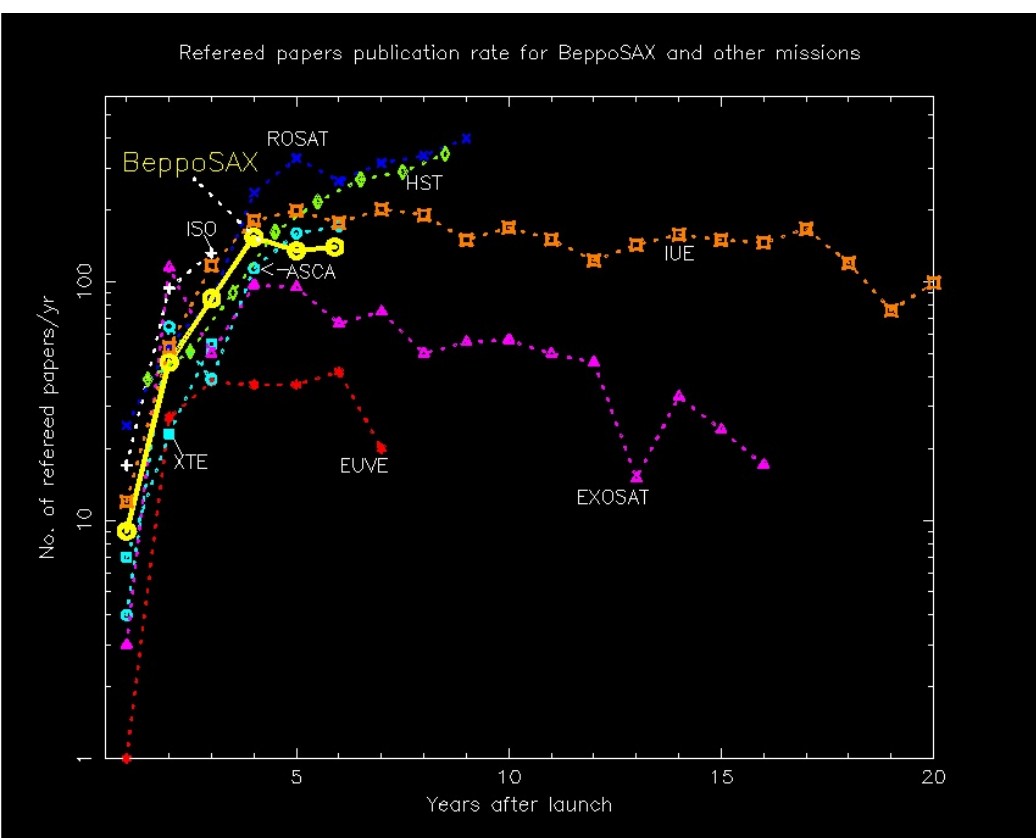

**Figure 15.** Number of citations of the papers based on the results with the *BeppoSAX* observations (mainly GRBs), compared with those based on the results obtained with other satellites. Figure kindly received by Paolo Giommi, at that time, the director of the ASI SDC (SAX Data Center).

### 6.2. Impact of the BeppoSAX Discoveries on GRB Theoretical Models

The measured distance scale of *BeppoSAX* GRBs swept away all the galactic models. The discovered properties of the events, like their huge isotropic energy (up to ∼$10^{54}$ erg), their non-thermal spectra, and their short time variability (down to ms time scale), were generally interpreted as a result of the formation of a fireball in relativistic expansion (see Figure 16). This model, already developed before the *BeppoSAX* dis-

covery of the X-ray afterglow (e.g., [56–58]), had immediate success for its capability to explain the spectral and temporal properties of the discovered GRBs (e.g., [39,59]), through the conversion of the fireball kinetic energy into electromagnetic radiation. This conversion was assumed to occur, for the prompt emission, through shocks between contiguous shells within the fireball, while, for the afterglow emission, through shocks in the external medium (e.g., [60,61]).

However, the smaller energy conversion efficiency in the internal shocks than in the external shocks was noted to be inconsistent with the observation results (e.g., [62]). Indeed, unlike the expectations from the fireball model, the energy released during the prompt emission was higher than that in the afterglow, at least on the basis of the afterglow spectrum, which was (and is still) possible to measure only up to 10 keV.

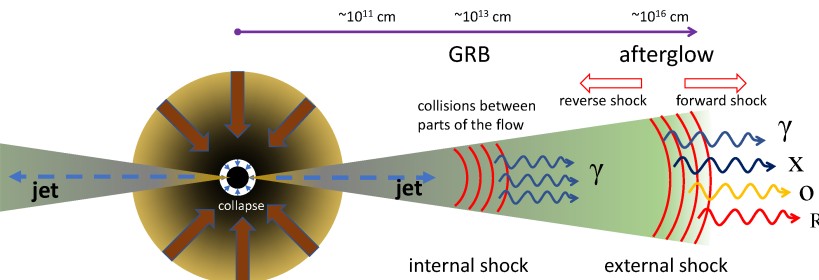

**Figure 16.** Sketch of a relativistically expanding fireball. The kinetic energy of the jet is converted through shocks (internal, external) in electromagnetic energy. Figure reprinted from [63].

Concerning the GRB progenitors, for short GRBs ($T_{90} < 2$ s), the merging of binary systems like white dwarf (WD)–neutron star (NS) or NS-NS or NS-BH, were considered the most likely mechanisms [64]. Instead, for long GRBs ($T_{90} > 2$ s), failed supernovae [65] or the collapse to a Kerr black hole of a rapidly rotating star (collapsar) [66] with the formation of hypernovae, were the most favorite models. Other models were also suggested for the GRB production, like the supranova model [67], or the transition, by accretion, of a neutron star to a quark star [68], or the Electromagnetic Black Hole (EMBH) model [69].

## 7. Other Relevant Results Obtained with *BeppoSAX*, with Most of Them Later Confirmed

Other relevant results concerning the prompt emission and the afterglow were obtained with *BeppoSAX*. Some of them, like the discovery of transient absorption lines and variable column density (see below), have still not been confirmed. This fact could be due to the passband of post-*BeppoSAX* wide field instruments, especially those that are more sensitive, whose lower energy threshold is in the hard X-ray band. The *THESEUS* mission concept [70] could definitely settle this issue. The most relevant *BeppoSAX* results are discussed in the following subsections.

### 7.1. Discovery of Transient Absorption Lines in the Prompt Emission

- A transient absorption edge at 3.8 keV in the prompt emission of the *BeppoSAX* GRB 990705 was first discovered [71]. The results are shown in Figure 17. The feature was found to be consistent with a red-shifted K-edge due to an iron environment. The confidence in the reality of this line is that the derived redshift ($z = 0.86$) was later measured from the GRB host galaxy [72].
- Investigating the spectral evolution of the prompt emission from the *BeppoSAX* GRB 011211 with measured redshift ($z = 2.140$), also evidence of a transient absorption feature at $6.9^{+0.6}_{-0.5}$ keV during the rise of the primary event [73] was found. The significance of the feature was derived with non-parametric tests and numerical simulations, finding a chance probability of $3 \times 10^{-3}$ down to $4 \times 10^{-4}$. The feature showed a Gaussian profile and an equivalent width of about 1 keV. See [73], where a possible interpretation is also discussed.

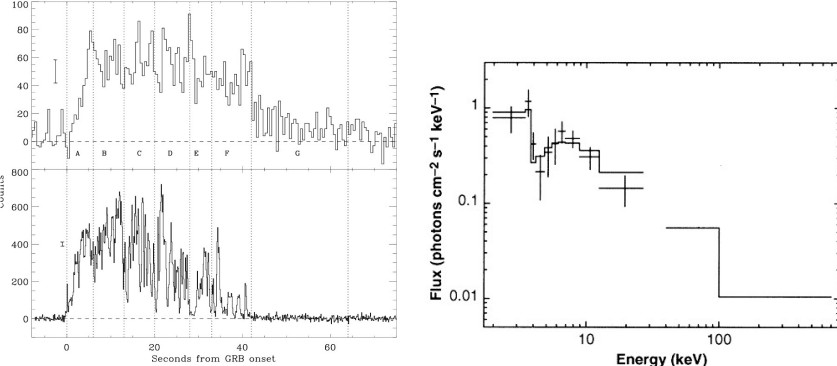

**Figure 17.** (**Top left**): light curve of GRB990705 obtained with *BeppoSAX* WFC (0.5 s time resolution). (**Bottom left**): light curve of GRB 990705 obtained with GRBM (0.128 s time resolution). The vertical dotted lines limit the 7 intervals, named A, B, C, etc, in which the spectral analysis was performed. (**Right**): photon spectrum of GRB 990705 in the time slice B. The best-fit curve is obtained with a power law plus a photoelectric absorption by a medium at redshift $z = 0.86$, column density $N_H = 1.3 \times 10^{22}$ cm$^{-2}$, and iron abundance 75 times the solar one. A similar fit was obtained for the spectrum of the time slice A, while the spectrum of the time slice C was well described by a simple power law. Figure reprinted from [71].

### 7.2. Detection of a Transient Column Density in the Prompt Emission

With *BeppoSAX*, thanks to the WFCs low energy threshold, also the detection of a decreasing column density during the prompt emission of GRB 000528 [74] was found (see Figure 18). This result is clear evidence that the GRB environment was ionized by the gamma-ray event.

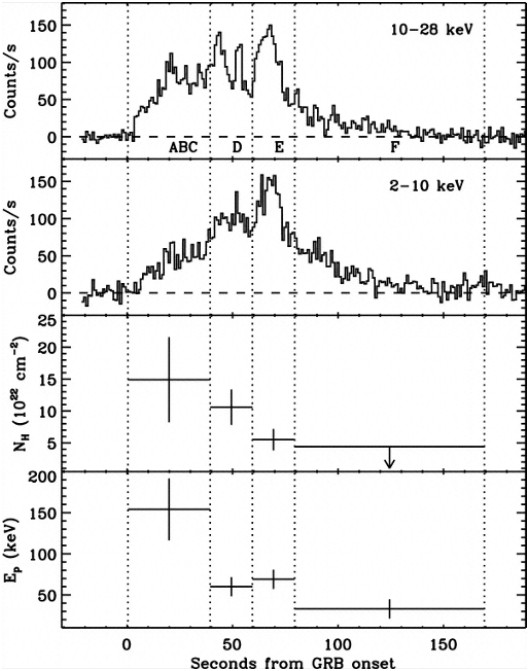

**Figure 18.** Measured column density as function of time from *BeppoSAX* GRB 000528 onset. Figure reprinted from [74].

Unfortunately, given the higher value of the low energy threshold of the later GRB monitors (at a minimum, 8 keV with *Swift* BAT and *Fermi* GBM), similar searches could not be performed until now. In an investigation performed using the *Swift* XRT data of 199 GRBs, 7 GRBs, in late time intervals (between about 1 and 2 min from the GRB onset),

showed signs of a decrease in N$_H$ (see review paper [75]). A definitive response to a ionization of the GRB environment could be given by the launch of the *THESEUS* mission concept (see Section 13).

### 7.3. Discovery of the GRB–Supernova Connection

For the first time, a *BeppoSAX* GRB (980425) was found to be the likely origin of a type Ic supernova (SN) explosion: SN1998bw [76]. The direction of this supernova had resulted to be coincident with that of GRB 980425, and its explosion was contemporary, within one day, with the occurrence of the GRB event. The SN was unusually bright (for this reason, it was called hypernova) and was expanding at a very high velocity [77].

The uncertainty about the chance coincidence of SN1998bw with GRB 980425 was definitively removed in 2003, when the type Ic SN2003dh was found to be associated with GRB 030329 [78]. Indeed, in this case, the spectra of the optical emission, initially consistent with a power-law continuum, after a week, became remarkably similar to those of SN1998bw, once they were corrected for the afterglow emission.

Nowadays, it is a matter of fact that several long GRBs originate in supernova explosions. In Table 1, we list the events with identified SNe, obtained by updating the list given by [79]. In addition to these identified SNs, many other GRBs have optical counterparts with an afterglow curve that exhibits a clear bump plus spectroscopic signatures typical of a SN (see review by [79]).

**Table 1.** List of GRBs associated with identified supernovae (SNe). ll = low-luminosity GRB ($L_{\gamma,iso} < 10^{48.5}$ erg s$^{-1}$), INT = intermediate-luminosity GRB ($10^{48.5} < \gamma, iso < 10^{49.5}$ erg s$^{-1}$). UL = ultra-long GRBs ($\sim 10^4$ s).

| GRB | Redshift, $z$ | GRB Type | SN Search |
|-----|-----|-----|-----|
| 980425 | 0.0085 | long, ll | 1998bw |
| 011121 | 0.362 | long | 2001ke |
| 021211 | 1.004 | long | 2002lt |
| 030329 | 0.16867 | long | 2003dh |
| 031203 | 0.10536 | long, ll | 2003nw |
| 050525A | 0.606 | long | 2005nc |
| 060218 | 0.03342 | long, ll | 2006aj |
| 081007 | 0.5295 | long | 2008hw |
| 091127 | 0.49044 | long | 2009nz |
| 100316D | 0.0592 | long, ll | 2010bh |
| 101219B | 0.55185 | long | 2010ma |
| 111209A | 0.67702 | UL | 2011kl |
| 120422A | 0.28253 | long | 2012bz |
| 120714B | 0.3984 | long, INT | 2012eb |
| 130215A | 0.597 | long | 2013ez |
| 130427A | 0.3399 | long | 2013cq |
| 130702A | 0.145 | long, INT | 2013dx |
| 130831A | 0.479 | long | 2013fu |
| 161219B | 0.1475 | long, INT | 2016jca |
| 171010A | 0.3285 | long | 2017htp |
| 171205A | 0.037 | long, ll | 2017iuk |
| 180728A | 0.117 | long | 2018fip |
| 190114C | 0.4245 | long | 2019jrj |
| 190829A | 0.07 | long, ll | 2019oyw |

In general, the associated SNe are type Ic with high expansion velocities and much larger energy release than in normal SNe, and thus are hypernovae as found in the case of SN1998bw. In addition, as can be seen from Table 1, most of them have a known redshift with values < 1.

However, there are GRBs with the same features of events associated with a SN (like their long duration, low luminosity, and redshift < 1) with no associated SN (see [80]),

demonstrating that there are GRBs originating in very faint supernovae or they are due to different phenomena.

### 7.4. Discovery of the $E_p$–$E_{iso}$ Relation

This relation, now also known as the *Amati relation* from the name of the first author of the discovery paper [81], gives the correlation between the photon energy, redshift-corrected, at the peak of the prompt spectrum $\nu F_\nu$ of a long GRB and the total energy released during the burst $E_{iso}$, assuming isotropy in the emission (see Figure 19). This relation was discovered with a set of 12 *BeppoSAX* long GRBs (>2 s), whose redshift was determined with optical spectrometers [81]. Two years later, after we published this result, two other relationships were reported: the "Yonetoku relation" between rest frame $E_{peak}$ and bolometric peak luminosity $L_{p,iso}$ [82], and the "Ghirlanda relation" between rest frame $E_{peak}$ and released energy $E_\gamma$ after correction for the beaming factor ($E_\gamma = (1 - cos\theta)E_{iso}$, where $\theta$ is the angular width of the jet), assuming a jet-like emission [83]. For a discussion on the weak points of the Ghirlanda relation, see [84].

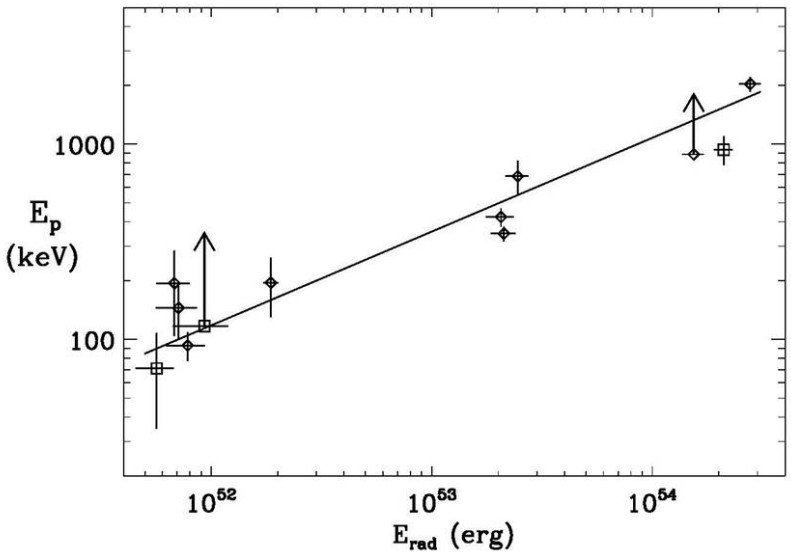

**Figure 19.** The $E_{p,i}$–$E_{iso}$ relation discovered with *BeppoSAX*. Reprinted from [81].

Also, relations between prompt and afterglow emission or about only the afterglow were reported. For a review, see [85].

Among these relations, the $E_p$–$E_{iso}$ relation remains the most robust. It is now confirmed that this relation is satisfied by almost all long GRBs, for which, along with $z$, it has been possible to estimate released energy and peak energy $E_p$. The only outliers are two long GRBs: 980425 associated with SN1998bw, and 031203 associated with SN2003nw (see Table 1). Possible explanations for these apparent outliers have been investigated. According to [86,87], from the fact that these two bursts share several properties with GRB 060218, an event associated with SN2006aj, which, however, obeys the $E_p$–$E_{iso}$ correlation, these authors suggest that such discrepancy could be due to the limited energy band in which these two events have been observed that could have biased the derived $E_p$ value.

Some authors suspected that the $E_p$–$E_{iso}$ relation could be influenced by selection effects (see [88]). However, by slicing the GRB time profiles in several time intervals and deriving the spectra in each of them, the correlation between the time resolved $E_p$ and the corresponding flux is still apparent (see [84,89,90]). A possible origin of the correlation was also discussed by [91].

The Amati relation also appears to be a promising tool to describe the expansion history of the universe and to estimate the cosmological parameters (see [92,93]). A recently

updated $E_p$–$E_{iso}$ relation is shown in Figure 20. The relation is discussed in the context of a review on cosmological probes [94].

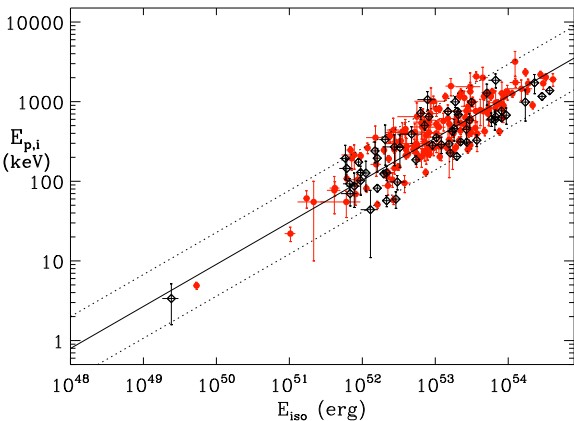

**Figure 20.** The $E_{p,i}$–$E_{iso}$ correlation for a recent sample of 224 long GRBs. Those events detected and localized by the *Swift* satellite are shown in red color, while those in black color were detected with other satellites (*BeppoSAX*, *INTEGRAL*, *Fermi*, etc). Courtesy of Lorenzo Amati.

### 7.5. Discovery of X-ray Flash and X-ray Rich Events

#### 7.5.1. X-ray Flash Events

X-ray flashes (XRFs) were discovered with *BeppoSAX* as bright, low-energy events with a duration like that of long GRBs. They were detected with the WFCs (2–28 keV) but undetected by GRBM (40–700 keV) [95]. Their temporal and spectral properties were found to be very similar to those of the X-ray counterparts of GRBs [96].

These events were later better investigated with the *HETE-2* satellite [97]) launched in October 2000, and then with the *Swift* satellite (see below). Thanks to *HETE-2* it was possible to establish (see [98]) that X-ray flashes show properties similar to those of long GRBs but with a lower peak energy $E_p$ (see top panel of Figure 21) and a lower redshift distribution (see bottom panel of Figure 21).

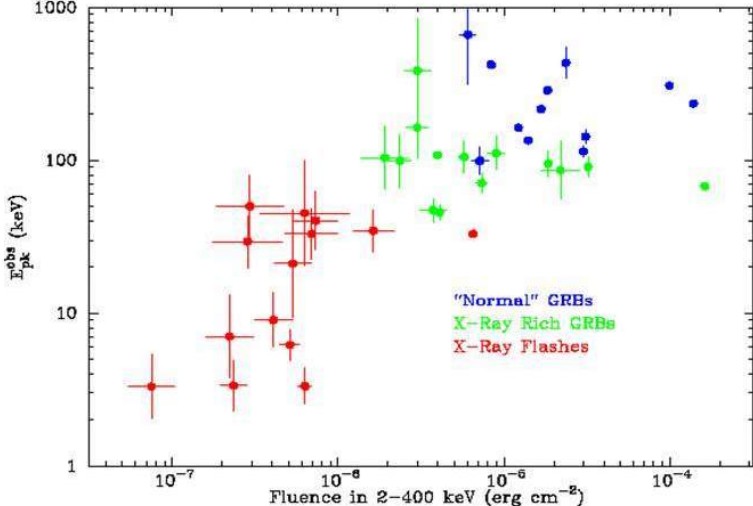

**Figure 21.** *Cont.*

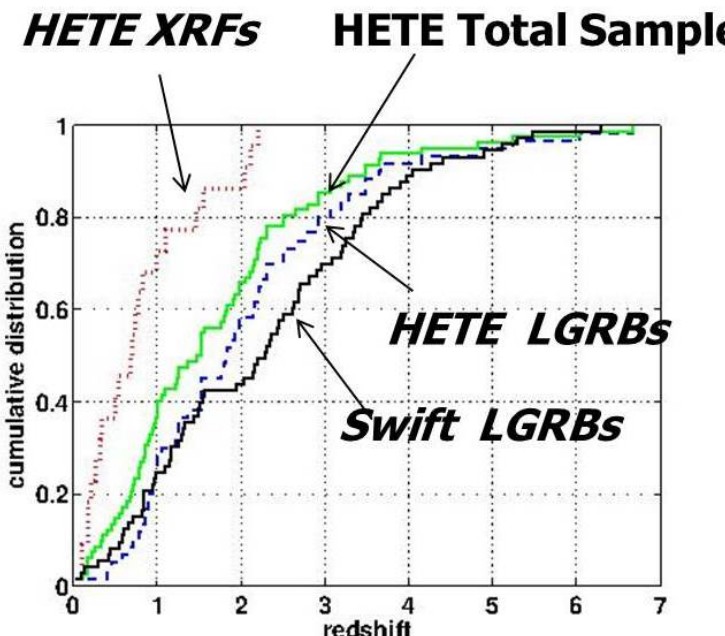

**Figure 21.** (**Top panel**): Distribution of the peak energy $E_p$ vs. fluence for XRFs, X-ray rich (XRR) events, and GRBs. Adapted from [99], where you can find a discussion on XRF and XRR events. (**Bottom panel**): redshift distribution of GRBs and XRFs. Figure adapted from that of [98].

XRFs were further investigated with *Swift*. They were found to have a $T_{90}$ duration between 10 and 200 s and an isotropic sky distribution. Thus, in these respects, XRFs are similar to "classical" GRBs. Also, the spectral analysis showed that their spectra were similar to those of long GRBs, with the main difference that XRFs have peak energies $E_p$ of their $EF(E)$ spectra that are much lower than those of long GRBs. Therefore, XRFs could be a subpopulation of GRBs with low peak energies.

An interesting discovery, in favor of the GRB-like picture, was the observation of an XRF event (060218) associated with the type Ic SN2006aj supernova [100]. SN2006aj was intrinsically less luminous than other previously discovered SNe associated with GRBs but more luminous than many supernovae not accompanied by a GRB. From the weakness of both the GRB output and the supernova radio flux, it could be established that XRF 060218 was an intrinsically weak and soft event rather than a classical GRB observed off-axis. These results extended the GRB–supernova connection to X-ray flashes and fainter supernovae, implying a common origin.

7.5.2. X-ray Rich Events

Also, X-ray rich GRBs (XRRs) were discovered with *BeppoSAX* [96]. Their properties were found to be intermediate between XRFs and classical GRBs.

The XRF and XRR interpretation as a subclass of classical GRBs is now strongly confirmed: see a recent review on the XRF properties by [101], in which it is definitely found that both XRF and XRR events are GRBs with the following main properties:

(a)  Low values of $E_p$, thus main fraction of energy released in the X-ray band;
(b)  Low luminosity;
(c)  Long duration;
(d)  $E_p$–$E_{iso}$ correlation as GRBs;
(e)  Same redshift distributions as GRBs;
(f)  The association with SN explosions is favored.

### 7.6. Spectral Properties of the Late Afterglow of Long GRBs

Concerning the late GRB afterglow, *BeppoSAX* also allowed to derive its average spectral properties with the LECS/MECS telescopes (0.2–10 keV). The average properties of the late afterglows were summarized as follows [102]:

- Photon spectra consistent with a photo-electrically absorbed power law, with photon indices $\alpha$ distributed according to a Gaussian with average photon index $\alpha_{ave} = 1.95 \pm 0.03$ and standard deviation $\sigma_\alpha = 0.4$.
- Fading behavior also consistent with a power law with index $\beta$ distributed according to a Gaussian ($\beta_{ave} = -1.30 \pm 0.02$; $\sigma_\beta = 0.35$).

With *Swift*, the late afterglow spectral properties have been fully confirmed, while the temporal properties, extended to the early afterglow, have been found to show a much more complex behavior (see Section 8.1).

About the hard X-ray band, with *BeppoSAX* it was also detected the first X-ray afterglow above 10 keV. This detection was obtained in the case of the very strong GRB 990123. In this case, it was possible to detect the afterglow spectrum up to 60 keV with the PDS instrument [103]. The high-energy spectrum was found not to be consistent with the power-law spectrum measured below 10 keV with LECS/MECS. An interpretation of the hard X-ray spectrum in terms of an Inverse Compton (IC) component in addition to a synchrotron spectral emission was also discussed [104].

After *BeppoSAX*, the hard X-ray afterglow spectrum was detected in only two cases of very bright GRBs. One case is the long GRB 120711A detected with the *INTEGRAL* satellite [105] for 10 ks in 20–40 keV, and, for a shorter time, at higher energies. According to these authors, likely the long duration of the high energy emission was still prompt emission. In any case, for this GRB, by combining the *INTEGRAL* IBIS and the *Fermi* LAT instruments, the hard X-ray energy spectrum was found consistent with synchrotron radiation.

Another case was that of the bright GRB 130427A observed with *NuSTAR* (3–79 keV) [106], starting approximately 1.2, 4.8, and 5.4 days after the *Swift* GBM trigger. In this case, it was shown that the hard X-ray observations (up to 80 keV) were crucial to establish that the afterglow was due to synchrotron radiation, and to provide a strong direct observational support for such an emission mechanism also in the high-energy gamma-ray band, investigated with the *Fermi* mission.

Hard X-ray afterglow emission up to about 20 keV was also measured with *NuSTAR* in the case of GRB 130925A [107], but given the narrow band of the GRB detection in hard X-rays, no definite conclusion about the origin of the hard X-ray component could be drawn.

### 7.7. BeppoSAX GRBM Catalog of GRBs

The *BeppoSAX* GRBM instrument detected 1082 GRBs (see catalog in [108]) with 40–700 keV fluences in the range from $1.3 \times 10^{-7}$ to $4.5 \times 10^{-4}$ erg cm$^{-2}$, with the discovery of very peculiar events, like that shown in Figure 22, featuring many peaks with subsecond duration and the finding that the GRB active time (i.e., the time $T_a$, in which the burst was visible above a $2\sigma$ level) has a cutoff at about 200 s, in spite of the fact that the found GRBs had a total duration up to 600 s. This result could be related to the fact that the inner engine could have a maximum active time of about 200 s.

Also, a spectral catalog was derived from the GRB detections [109], in which the average spectra of the 200 brightest GRBs were reported and are publicly available. Most of the photon spectra were found to be consistent with a Band function, with low-energy index around 1.0 and a peak energy $E_p$ of the $EF(E)$ spectrum of about 240 keV, in agreement with previous results obtained on bright GRBs with the *CGRO* BATSE experiment. It was interesting to see the fact that about 30% of the GRB spectra could be fit with a simple power law, suggesting that the peak energy could be close to or outside the GRBM energy boundaries as later extensively confirmed with broader passband missions, e.g., *Fermi*.

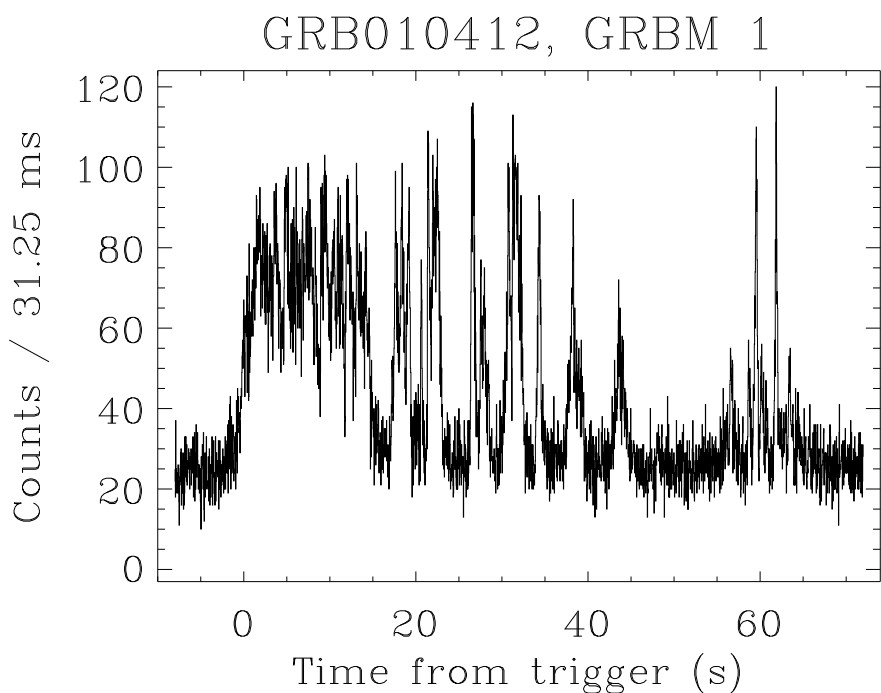

**Figure 22.** GRB 010412 as detected by GRBM aboard *BeppoSAX*, with 7.8 ms time resolution.

## 8. New Discoveries on GRBs in the Post-*BeppoSAX* Era

In addition to the confirmation of the huge discoveries obtained with *BeppoSAX*, many questions left unanswered by this satellite were solved in the post-*BeppoSAX* era, like the early afterglow properties, the breaks in the X-ray afterglow light curves, the afterglow of short bursts and their origin, and the GRB environment.

### 8.1. Swift and Its Role in the Progress of the GRB Phenomenon

As already mentioned, the *Swift* satellite [52], launched on 20 November 2004 and still operational at this time, is one of the post-*BeppoSAX* missions that has already given and is still giving a very high contribution to the GRB astrophysics. The payload configuration is similar to that of *BeppoSAX* with a hard X-ray (15–150 keV) wide field (1.5 sr) BAT telescope for GRB detection and localization, and a focusing XRT telescope (0.1–10 keV) for the afterglow measurement and study. Thanks to its autonomous slew decision capability, *Swift* can start the afterglow observations in less than 2 min time. A further *Swift* performance is due to a UV–Optical Telescope (UVOT) on board, that allows to identify the optical counterparts of the X-ray afterglow sources with a limiting magnitude, in 1000 s, of $B = 22.3$ in white light.

Thanks to this configuration, its rapid reaction capability and its long life time (already about 20 years), several new results have already been obtained with this mission, with reviews of the most important ones, inclusive of those obtained from statistical studies, already reported (see [110–119]). Among them, two important results, left open by *BeppoSAX* and solved with *Swift* thanks to its rapid reaction capability, are the early afterglow properties of long GRBs and the afterglow of short GRBs (no afterglow of short GRBs was discovered with *BeppoSAX*).

The afterglow detection of short GRBs allowed to determine their optical counterparts and thus their distances with the consequent determination of the energy released in these events, their redshift distribution and other properties, among them being the discovery that short GRBs are outliers of the Amati relation (see Figure 23), forming a similar relation offset from the main correlation.

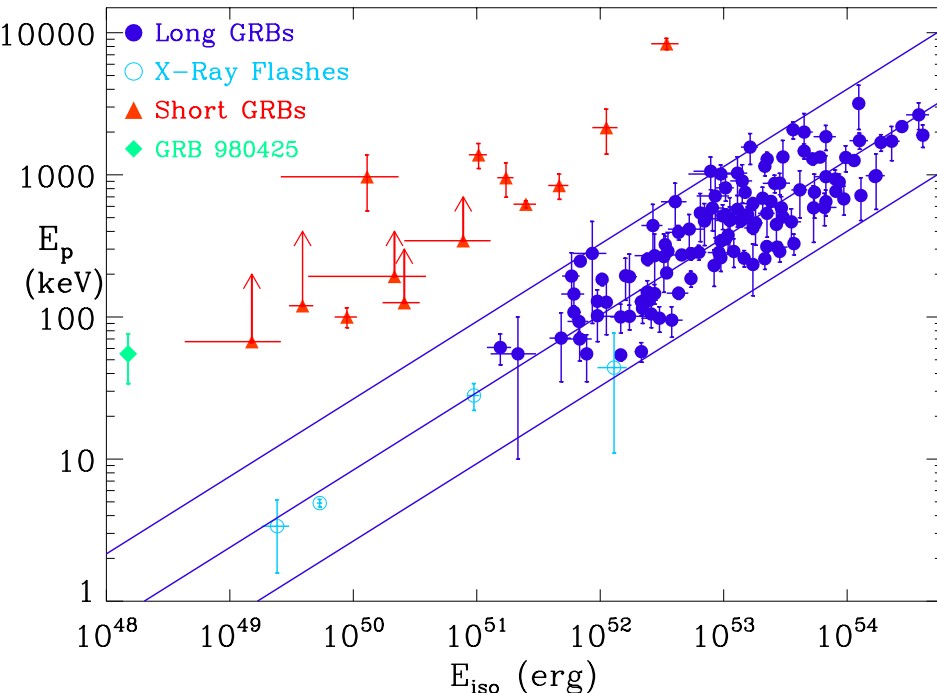

**Figure 23.** The $E_p$–$E_{iso}$ relation in which short GRBs are also reported. As can be seen, short GRBs are outliers of the Amati relation but form a similar relation. Courtesy of Lorenzo Amati.

Concerning the afterglow behavior from early to late times, a first study based on 27 long GRBs [120] observed with *Swift* showed the presence of a canonical light curve, with three distinct power-law decays ($\propto t^{-\beta}$): initially ($\lesssim 500$ s) very steep ($3 \lesssim \beta \lesssim 5$), then ($10^3 \lesssim t \lesssim 10^4$ s) very shallow ($0.5 \lesssim \beta \lesssim 1$), and finally, after several hours, a decay with $1.0 \lesssim \beta \lesssim 1.5$, as found with *BeppoSAX*. With a much more significant sample of *Swift* GRBs (622 long, and 36 short), the afterglow properties were deeply investigated by [121], inclusive of a statistical analysis of a possible relation of the X-ray afterglow properties with the prompt $\gamma$-ray emission. Their results showed that, unlike what was initially found [120], the light curves exhibit a variety of power-law slopes (see Figure 24), with, in some cases, X-ray flares during the early afterglow, superimposed to smoothed power laws (see Figure 25).

Among the many other results, thanks to the *Swift* configuration and its prompt dissemination of the GRB locations, it merits mentioning the possibility allowed by *Swift* to perform a prompt optical and radio follow-up, with the determination of the redshift of a significant number of optical/radio GRB counterparts (see Figure 26). As can be seen from the figure, the redshift distribution extends up to high $z$, with the highest redshift events being GRB 090423 with $z = 8.26$ and GRB 090429B with $z = 9.4$.

Studies of the GRB distribution with $z$ (see [122]) have shown a significant correlation between the density of the GRB rate, with redshift between 0 and 4, and star formation rate (SFR) density in the same redshift interval. The ratio $\Psi(z)$ between the GRB rate and SFR is found to follow a power law ($\Psi(z) \propto (1+z)^{0.5}$) which can be explained if GRBs occur in low-metallicity galaxies. If this relation would continue for $z > 4$, the discovery rate of very distant GRBs would imply a SFR density much higher than that inferred from UV-selected galaxies. The discovery of a significant sample of high $z$ GRBs is still not obtained and is expected to be obtained, e.g., by the mission concept *THESEUS*.

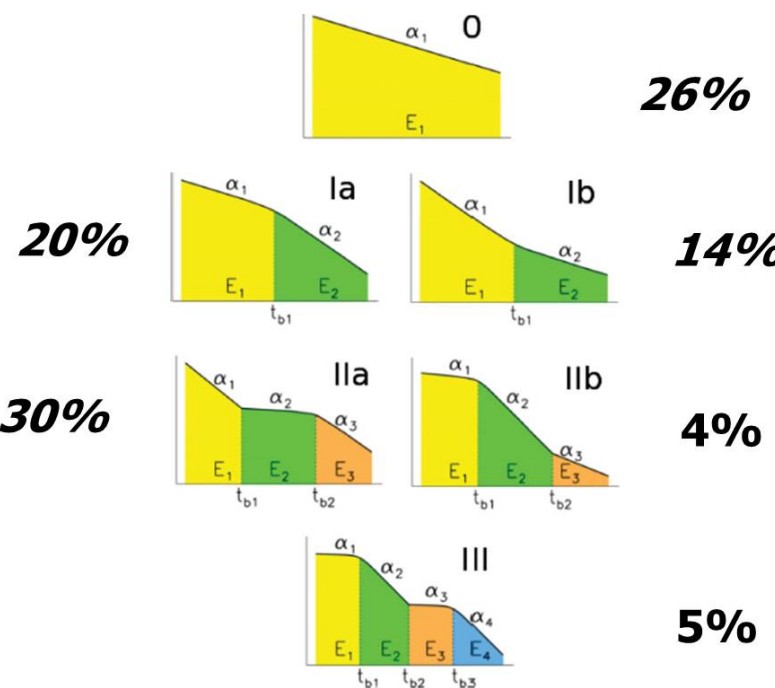

**Figure 24.** Different types of afterglow light curves in 0.1–10 keV observed with *Swift* starting from $t \gtrsim 60$ s. Also, the fraction of GRBs with the given type is shown. Initially, it seemed that the canonical light curves were either the IIa, whose frequency is only 30% of the times, or the Ia, whose frequency is 20%. Figure adapted from that of [121].

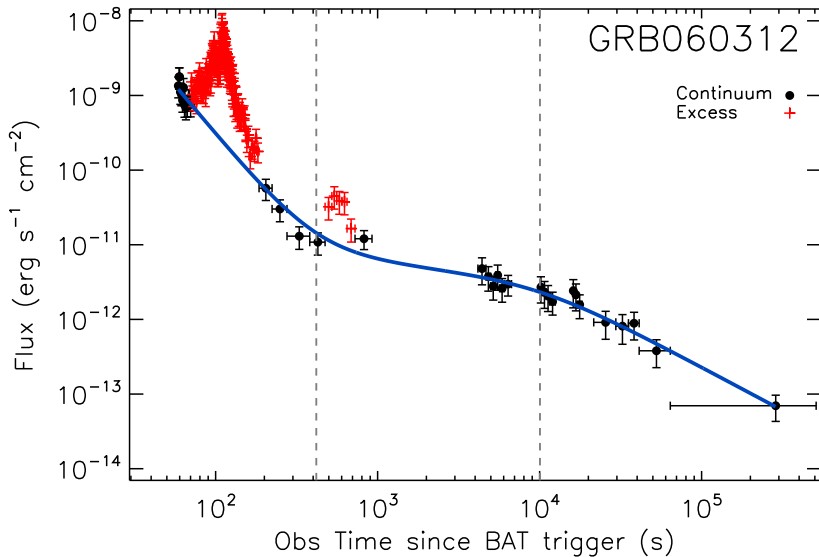

**Figure 25.** Example of light curve of the intrinsic 0.6–30 keV afterglow starting from $T \geq 60$ s as detected with *Swift*. Two X-ray flares above the smoothed light curve are apparent. Reprinted from [121].

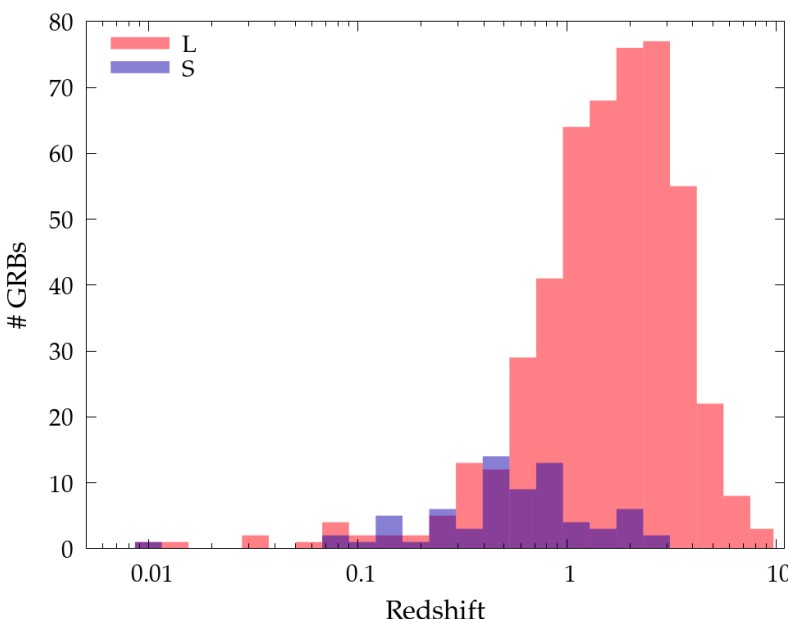

**Figure 26.** Redshift distribution of 70 short (purple) and 488 long GRBs, list updated up to April 2024. Most of them have been discovered and promptly localized with *Swift*. The smaller distance scale of short GRBs is clear. Figure kindly provided by Cristiano Guidorzi.

### 8.2. The Contribution to GRB Studies by High-Energy Gamma-Ray Space Missions and VHE Ground Telescopes

Also, the already mentioned high-energy missions *AGILE* (30 MeV– 50 GeV), launched on 23 April 2007 and de-orbited in February 2024, and *Fermi* (20 MeV–300 GeV), launched on 11 June 2008 and still operational, have provided a great contribution to the understanding of the GRB phenomenon at high energies (see [123]). Among the most relevant results, it certainly merits mentioning the discovery that the onset of the GRB high-energy gamma-ray (>100 MeV) emission is delayed with respect to that at lower energies (see left panel of Figure 27). Another discovered feature is that the gamma-ray spectrum hardens with time from the GRB onset, and a further high-energy spectral component appears in the tail of the gamma-ray light curve (see right panel of Figure 27). This behavior is confirmed also in the case of very bright events (see e.g., GRB 130427A [124]). All that shows that the high-energy gamma-rays arise from a region and/or a mechanism which is different from that which produces the lower-energy photons. This property is in agreement with the standard model of GRBs, in which the blast wave that produces the initial, bright prompt emission later collides with the circumburst medium, creating shocks. These external shocks can accelerate the charged particles and produce photons through synchrotron radiation. The observed high-energy photon energy distribution is consistent with this picture (see e.g., [106]).

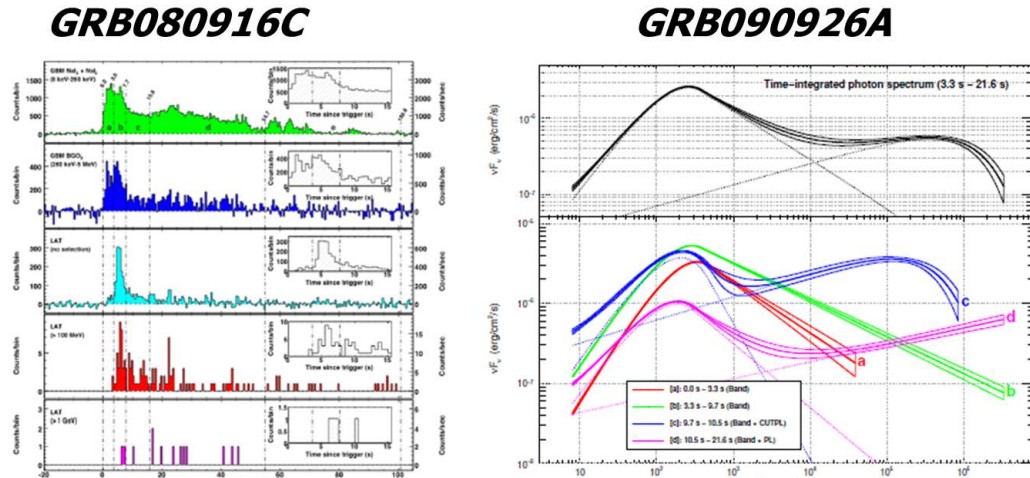

**Figure 27.** (**Left panel**): Light curve of the *Fermi* GRB 080916C at different energies. The delay of the onset of the high energy gamma-rays is apparent with respect to the onset of the GRB at low energies. Adapted from [125]. (**Right panel**): Time-resolved spectra of the *Fermi* GRB 090926A. A hardening of the high-energy spectrum with time is apparent when the high-energy gamma-ray spectral component appears during the latest time interval. Reprinted from [126] © AAS. Reproduced with permission.

An important test for establishing whether the afterglow emission is still due to synchrotron at Very High Energies (VHE, > 100 GeV) can be performed with Cerenkov telescopes. To date, only a few GRBs have been detected at these energies: 180720B [127], 190114C [128], and 190829A [129]. From a multi-wavelength spectral analysis performed by [130], it was found that the VHE emission (0.1–1 TeV) detected from GRB 190114C, with the *Major Atmospheric Gamma-ray Imaging Cherenkov* (*MAGIC*) telescopes [128], is very likely due to the inverse Compton up-scattering of synchrotron photons. However, the VHE afterglow emission (0.18–3 TeV) detected from GRB 190829 with the High Energy Stereoscopic System (*H.E.S.S.*) is consistent with synchrotron radiation [129].

## 9. GRB Progenitors in the Post-*BeppoSAX* Era

### 9.1. Long GRBs

For long GRBs ($T_{90} > 2$ s), there is still a general consensus that they are mostly due to the core collapse of very massive stars (dubbed as "collapsars", see [65]). This conclusion is justified from the following facts: (a) the well-established GRB-SN connection; and (b) long GRBs are located in the brightest regions of high SFR galaxies, where the most massive stars are located. However, recently [131,132], a long GRB (211211A) was associated with a kilonova whose progenitor is a compact binary merger, suggesting that GRBs with long, complex light curves can be spawned also from merger events.

A different origin for the progenitors of long GRBs was proposed by Ruffini et al. [133] in the case of long XRFs/GRBs, which exhibit two distinct episodes in their light curves. According to these authors, in these cases, a carbon–oxygen core undergoes a SN explosion, which triggers a hypercritical accretion onto a NS companion in a tight or more separated binary system. Depending on the system tightness, the formation or not of a BH is driven, and a GRB or an XRF event, respectively, is produced. A candidate for the formation of a black hole, according to these authors, was GRB 090618 ($E_{iso} = 3 \times 10^{53}$ erg), for which there was an evidence of an SN bump 10 days after the event [134].

### 9.2. Short GRBs

Concerning short GRBs, from the absence of evidence of simultaneous SN explosions and from the association with galaxies with a wide range of star formation properties (inclusive of low SFR), there is a general consensus that they are mostly the result of

compact binary (e.g., NS-NS, BH-NS) merging as it was initially supposed. The association of a gravitational wave signal (GW 170817) with a short GRB (170817) has confirmed this scenario [135]. However, also recently, some significant exceptions have been found. One is the case of the short, low-redshift GRB 200826A [136,137] that has been found to be associated with a SN from the optical and NIR bump in the light curve, and with luminosity and evolution in agreement with several SNe associated with long GRBs. Also, the prompt emission of this event follows the Ep,i–Eiso relation found for long GRBs.

From these last discoveries, it is emerging the opportunity to abandon the classification of short and long GRBs and to introduce a new classification: Type I and Type II GRBs. A useful discussion on this subject can be found in [70].

## 10. Physics behind GRB Events in the Post-*BeppoSAX* Era: Inner Engine

On the basis of the numerous observations of GRBs already made and the study of their properties, the physics of these events and their afterglows are now at an advanced level of knowledge, with very comprehensive reviews on these subjects (see e.g., [111] and references therein). As discussed in the cited review, there is a general agreement that GRBs are bright relativistic jets, while the GRB central engine, i.e., the mechanism by which this energy is released, is expected to be similar for long and short GRBs, with two possible main solutions proposed:

- One central engine mechanism assumes that GRBs are powered by mass accretion onto a stellar mass black hole at a very high rate (from a fraction to a few solar masses per second). In this case, the plasma is extremely hot and forms a thick disk or torus around the central black hole, from which a GRB jet is launched via three possible mechanisms: (1) a neutrino dominated accretion flow [138]; (2) extraction of electromagnetic energy by rotation of the black hole via a Poynting flux (mechanism also called Blandford–Znajek process) [139]; and (3) for the case of a highly magnetized accretion disk, the accumulation of energy with the launch of magnetic blobs from the differential accretion of the disk [140].
- The other central engine mechanism assumes that a GRB is powered by a rapidly spinning ($P_0 \sim 1$ ms period), highly magnetized ($B_{surface} \sim 10^{15}$ Gauss) neutron star ("fast magnetar") when it is spinning down. In this case, the maximum energy that can be extracted by the magnetar is given by [111]

$$E_{rot} \approx \frac{1}{2} I \Omega^2 \approx 2 \times 10^{52} \frac{M}{1.4 M_\odot} \left(\frac{R}{10^6 \text{cm}}\right)^2 \left(\frac{P_0}{1 \text{ ms}}\right)^{-2} \text{ erg}$$

If the energy released by a GRB is higher than that given by $E_{rot}$, the fast magnetar mechanism can be ruled out.

It is now well established that most GRB events are the results of the superposition of pulses. Figure 22 is a clear example of this property. Deep studies of the temporal and spectral properties of these pulses and their distribution and evolution are expected to help to clarify the engine inner mechanisms that give rise to the GRB jets [141,142].

A strong contribution to the study of the jet properties of GRBs and their inner engines has been also given by the observations of very strong events, in particular, from the brightest-of-all-time GRB 221009 recently observed (see e.g., [143,144]). This event is very unique, with a huge amount of very-high-energy photons (from 100 GeV to TeV) detected with LHAASO observatory [145]. The combined analysis of these data with those detected at lower energies is expected to be of key importance to obtain information on the dissipation process that gives rise to a GRB [145].

## 11. GRBs in the Multi-messenger Era

Until a decade ago, electromagnetic radiation was the only messenger of the phenomena occurring in the universe. Thanks to the enormous technological and financial efforts, in the last decade, gravitational waves (GWs), high-energy neutrinos (>10 TeV)

and ultra-high-energy cosmic rays ($>10^{18}$ eV) have begun to be messengers of astrophysical phenomena, with the birth of the multi-messenger astrophysics (see historical review by [146]). Evidence of neutrino emission has been detected from blazars [147], and (a very suggestive result!) a gravitational wave event (GW 170817) observed with the laser interferometric gravitational wave LIGO/VIRGO detectors has been associated with a short GRB event (GRB 170817A) [135], being almost simultaneously and independently detected. The localization of these GW and GRB events (see Figure 28) was not so constraining, but their near simultaneity, with the GRB event occurring only 1.7 s later, was fully consistent with the expectations for the possible production of a GRB event in the case of a NS-NS merging. Indeed, from the observed properties of the GW event, it was possible to establish that this event was due to the merging of a binary neutron star system with the NS mass in the range from 0.86 to 2.26 $M_\odot$ at a distance of $40^{+8}_{-8}$ Mpc. Also, the later optical and infrared discovery in the error box region of a kilonova (see [148]), i.e., an outflow rich of high atomic number nuclear elements as a result of the so-called r-process, confirmed this association.

After GW 170817, no more GW events have been found to be associated with GRB events, likely due to the fact that almost all of the GW events discovered thus far have been found to be the result of the merging of stellar-mass binary black holes, which are not expected to give rise to electromagnetic radiation. With the development of more sensitive gravitational detectors, like *Einstein Telescope* [149], the discovery of gravitational signals due to NS-NS merging will be much more likely. Binary black hole mergers are also expected in Active Galactic Nuclei [150], in which X-ray and gamma-ray radiation is expected to be emitted.

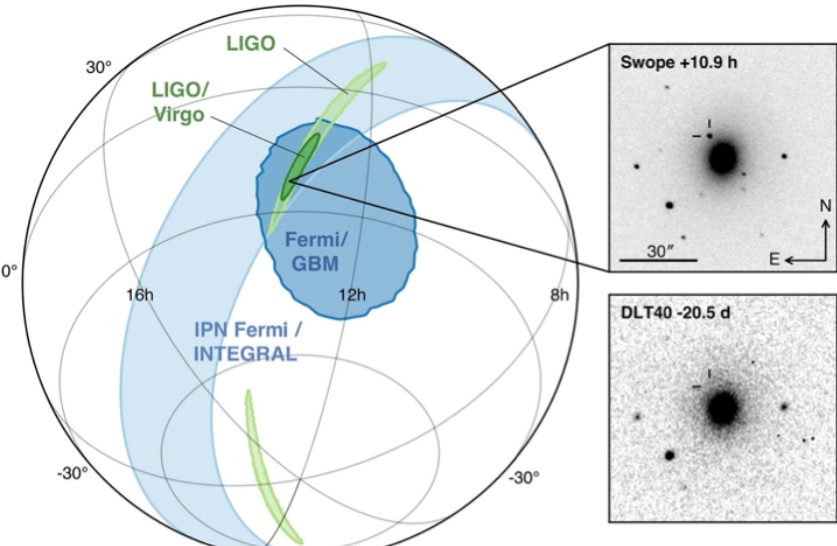

**Figure 28.** Error box in the localization of the gravitational event GW 170817 by the LIGO/Virgo detectors and that of the short GRB 170817A by *Fermi* GBM detector. Reprinted from the review by [135].

## 12. The Future for GRBs

### 12.1. Still Open Issues on GRBs

Several questions are still open regarding GRB physics and properties. As above discussed, one still open issue is the central engine (or engines) that powers the GRB events. As discussed by [111], given their observed non-thermal spectra, GRBs are expected to be efficient cosmic ray accelerators, with the first-order scenario being the Fermi acceleration mechanism in relativistic shocks (internal and external), alternatively a magnetic reconnection mechanism. The maximum energy of the accelerated protons can achieve Ultra-High

Energies (UHEs), whose interaction can produce neutrinos, in addition to gamma-ray photons. These neutrinos are being searched with still negative results (see [151]).

An interesting open issue is the determination of the hard X-ray afterglow spectrum. Up to 10 keV, as we have seen, the spectrum is well known and can be described by a power-law shape, consistent with synchrotron radiation. The question is, which is the higher-energy spectrum? an extrapolation of the lower-energy one, and thus can it still be considered synchrotron radiation? As we have already discussed in Section 7.6, to date, only for three brightest events, significant measurements of the high-energy spectrum of the GRB afterglow have been obtained. The accurate determination of the hard X-ray afterglow spectrum for a large sample of GRBs with its extension to higher energies is needed in order to better infer the prevailing emission mechanism at work.

### 12.2. Role of GRBs for Cosmology and Fundamental Physics

Given their huge brightness, GRBs can be detected up to the highest redshifts as already demonstrated with *Swift* GRBs (see Section 8.1). This allows to extend the Hubble diagram substantially beyond the redshift range of Type-Ia SNe and thus to further test the standard cosmology $\Lambda$CDM and its cosmological parameters (see [94]).

GRBs are also crucial for understanding the formation of the first collapsed structures (Pop-III and early Pop-II stars) and, thus, how and when the first stars formed and how they influence their environment and thus the re-ionization of the intergalactic medium.

Given their cosmological distance, GRBs also offer the opportunity to settle several questions of fundamental physics. Among them are the following:

- Test of the Lorentz invariance violation that is expected in some theories of quantum gravity. This test can be performed by measuring the time delay of photons as a function of energy.
- Investigation of the BH physics through signatures in the timing properties of GRB prompt emission.
- As also demonstrated by the gravitational wave event GW 170817 associated to GRB 170817A, these types of events provide the unique opportunity to study theories of gravity also beyond general relativity (see [152]).
- Given that, as discussed above, GRBs are the result of ultra-relativistic shocks with Lorentz factors of several hundreds, much higher than other possible accelerators like blazars, GRBs are believed to be strong candidates for particle acceleration to extreme energies that are currently observed among cosmic rays. From the study of their spectrum, it is possible to constrain the energy distribution of such accelerated particles. It has also been proposed that GRB reverse shocks may serve as potential accelerators of ultra-high-energy cosmic rays [153].
- GRBs are also strong candidates to contribute to the observed UHECRs and high-energy neutrinos because of the extreme shock acceleration caused by the newborn compact object. An important role in the production of high-energy neutrinos is expected by low-luminosity GRBs (see [154] and references therein).
- Another open issue of fundamental physics is the existence of axion-like particles (ALPs) and their oscillation with photons. Given their cosmological distance, GRBs could provide a tool for testing this issue from the observation of high-energy gamma-ray photons arriving from very high distances. The recent detection of photons at the energy of several TeV from the GRB221009A ($z = 0.151$) seems to be in contradiction with the expected optical depth for electromagnetic radiation, and has been recently interpreted in terms of the existence of ALPs, with a mass in the range from $10^{-11}$ to $10^{-7}$ eV, that oscillate with photons [155].

GRBs are unique events capable of producing peculiar and well-distinguished electromagnetic radiation up to the highest frequencies that can be detected also from very large distances.

**13. Future Space Missions and Ground Facilities Devoted to GRBs**

The still open issues on GRBs and those on fundamental physics that are expected to be solved by GRBs have motivated the development of new space missions and ground facilities.

*13.1. Space Missions*

- A mission that has the potentiality of detecting GRBs in the X-ray energy band, where they are less explored, is the Chinese mission with an international participation *Einstein Probe* (*EP*), that has been very recently launched (9 January 2024). It has on board two instruments: (1) a Wide-field X-ray Telescope (WXT), based on lobster eye optics, with a large FOV (60 × 60 deg) for transient source survey and localization, and a passband from 0.5 to 4 keV; and (2) a Follow-up X-ray Telescope (FXT), with two units, each one with Wolter I focusing optics, a focal length of 1.6 m, a narrow FOV (1 × 1 deg) and an energy passband 0.2–10 keV [156]. Given the low-energy band of its instruments, the association of an *EP* transient event with a GRB requires the simultaneous detection of the event hard X-ray/soft gamma-ray emission.
- An X-ray/gamma-ray space mission, scheduled to be launched on 24 June 2024, is the Chinese–French mission *SVOM* that has on board a Gamma-ray Monitor (15 keV–5 MeV), an X-ray imager and trigger (ECLAIRs, 4–150 keV), a lobster eye telescope (MXT, 0.2–10 keV) and an optical telescope [157].
- An already mentioned mission concept for its importance is *THESEUS* [70], that has been approved by ESA for a new phase A study. If it will be adopted, the launch is expected to be in 2035. It has on board three instruments, two with a wide FOV (a Soft X-ray Imager (SXI, 0.3–5 keV), based on a lobster eye focusing system, and a broad energy band (2 keV–10 MeV) X-ray Gamma-ray Imaging Spectrometer (XGIS) for GRB identification and accurate localization), and a 70 cm class InfraRed Telescope (IRT, 0.7–1.8 μm) with imaging and spectroscopic capabilities (resolving power, $R \approx 400$, through $2' \times 2'$ grism), for the GRB IR counterpart identification and its redshift determination.
- In the gravitational wave field, a large mission has been recently approved: the ESA-NASA mission *eLISA* (Laser Interferometer Space Antenna), foreseen to be launched in the early 2035s [158].

*13.2. Ground Facilities*

Extremely large optical, radio, neutrino, gravitational wave facilities are already operational or under development, to extend the band of GRB detection and their multi-messenger counterparts.

- Among the future optical facilities, I mention the largest ones: the European Extremely Large Telescope *EELT* (see [159]), the US Thirty Meter Telescope *TMT* (see [160], the American Giant Magellan Telescope (*GMT*) (see [161]) and the Vera Rubin Observatory [162]. In particular, the latter will execute a Legacy Survey of Space and Time (LSST) of the entire southern sky every four nights in six different bands during ten years.
- Also, new radio facilities are under development, particularly the Square Kilometer Array *SKA*, that will be the largest radio telescope in the world (see [163]).
- In the gravitational wave field, a gravitational wave European project is being developed (location still not decided): the *Einstein Telescope* (see [164]).
- In the Very-High-Energy (VHE) gamma-ray field, there are already several operational facilities, like *MAGIC* (see [165]) and *HESS* (see [166]) already seen, *VERITAS* (Very Energetic Radiation Imaging Telescope Array System; see [167]), and *LHAASO* (Large High Altitude Air Shower Observatory; see [168]). In addition, a very large project is under development: *CTA* (Cerenkov Telescope Array; see [169]).
- Also, large neutrino facilities are already operational, like the underwater telescope *ANTARES* (Astronomy with a Neutrino Telescope and Abyss environmental RESearch project; see [170]), and the *ICECUBE* neutrino observatory (see [171]). Instead, a

next-generation neutrino facility is the telescope *KM3NET* (Cubic Kilometre Neutrino Telescope; see [172]).

## 14. Concluding Soft Gamma-Ray Mission Concept

I would like to conclude my review of the future space missions for deep studies of GRBs by mentioning an advanced X-ray/soft gamma-ray mission concept, *ASTENA* (Advanced Surveyor of Transient Events and Nuclear Astrophysics) under study by an international collaboration led by the University of Ferrara in the context of the European program *AHEAD*. The foreseen payload (see Figure 29) includes two main instruments: an array of 12 Wide Field Monitor-Imager Spectrometers (WFM-IS) with a passband from 2 keV to 20 MeV, and a focusing Narrow Field Telescope (NFT) with a passband from 50 to 700 keV, based on a Laue lens of about 7 m$^2$ collecting area and a 20 m focal length [173].

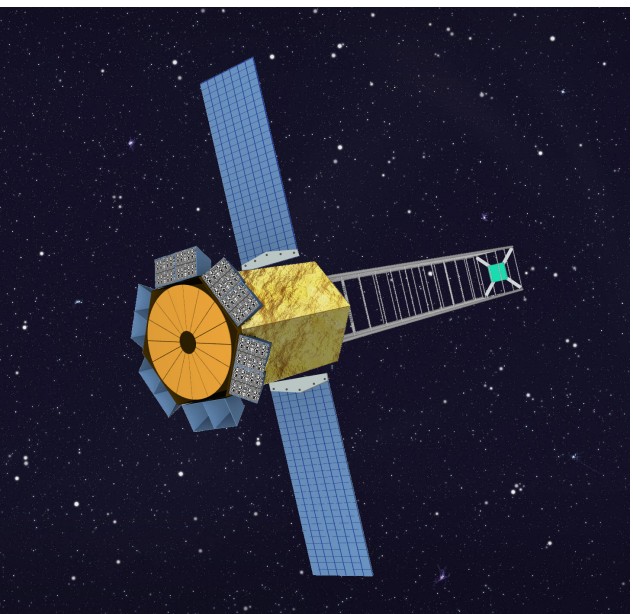

**Figure 29.** The *ASTENA* mission concept, proposed to ESA for its long-term program "Voyage 2050". Reprinted from [173].

Thanks to its large effective area (5800 cm$^2$ at energies <30 keV and larger at higher energies), its wide FOV (2 sr) and very good angular resolution (6 arcmin), WFM-IS could perform deep studies of the transient sky, particularly low-luminosity GRBs and other still unknown high-energy transients, contributing so much to future multi-messenger astronomy [174].

On the other side, NFT is expected to improve by orders of magnitude the sensitivity of the best current gamma-ray instruments, inclusive of NuSTAR. NFT will be ideal for studying the still almost unknown soft gamma-ray spectra of GRB afterglows.

Two white papers, based on the *ASTENA* mission concept, were submitted to ESA for its long-term program "Voyage 2050" [173,174]. The discussed topics were recommended from the Voyage 2050 Senior Committee for a medium size satellite mission.

All these space and ground facilities are expected to provide further breakthrough results for the GRB astrophysics and, more generally, for continuing in the advancement of our knowledge of the Universe.

**Funding:** This research received no external funding.

**Acknowledgments:** Many people contributed to the *BeppoSAX* discoveries. I wish to thank all of them. I wish to thank the book editors and the three anonymous referees for their very useful suggestions and contributions to improve the paper. Many thanks to Mauro Orlandini for his help in

different aspects of the paper preparation. Lastly, many thanks to my wife for her patience during my days of work at home for this review, when I should have been available for family needs.

**Conflicts of Interest:** The authors declare no conflict of interest.

**Abbreviations**

The following abbreviations are used in this manuscript:

| | |
|---|---|
| ASI | Italian Space Agency |
| ASTENA | Advanced Surveyor of Transient Events and Nuclear Astrophysics |
| BAT | Burst Alert Telescope, aboard *Swift* |
| BATSE | Burst and Transient Source Experiment |
| BH | Black Hole |
| CGRO | Compton Gamma Ray Observatory |
| FOV | Field of View |
| FWHM | Full Width at Half Maximum |
| GRB | Gamma-Ray Burst |
| HPGSPC | High-Pressure Gas Scintillator Proportional Chamber, aboard *BeppoSAX* |
| HRI | High-Resolution Imager aboard *ROSAT* |
| IAU | International Astronomical Union |
| IBIS | Imager on Board Integral Satellite |
| LAD | Large Area Detectors of the BATSE experiment |
| LECS | Low Energy Concentrator Spectrometer, aboard *BeppoSAX* |
| MECS | Medium Energy Concentrator Spectrometer, aboard *BeppoSAX* |
| NASA | National Aeronautics and Space Administration |
| NFIs | Narrow Field Instruments aboard *BeppoSAX* |
| NIR | NearInfraRed band |
| NS | Neutron Star |
| PI | Principal Investigator |
| PDS | Phoswich Detection System, aboard *BeppoSAX* |
| ROSAT | ROengten SATellite |
| SAX | Satellite Astronomia X (X-ray Astronomy Satellite in Italian) |
| SDC | SAX Data Center |
| SFR | Star Formation Rate |
| SRON | Space Research Of Netherlands |
| THESEUS | Transient High-Energy Sky and Early Universe Surveyor |
| UVOT | Ultraviolet/Optical Telescope, aboard *Swift* |
| WD | White Dwarf |
| WFCs | Wide Field Cameras, aboard *BeppoSAX* |
| XRF | X-Ray Flash |
| XRR | X-Ray Rich GRB |
| XRT | X-Ray Telescope, aboard *Swift* |

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
