# Peer review of "A Short History of the First 50 Years: From the GRB Prompt Emission and Afterglow Discoveries to the Multimessenger Era"

_universe, doi:10.3390/universe10060260_

Round 1
Reviewer 1 Report
Comments and Suggestions for Authors
Dear Filippo,
Thanks for producing this valuable walk-through of the GRB discovery timeline. I read your paper and found it genuinely exciting and pleasurable to read. I indicate here some improvements and suggestions, mostly rephrasing. I come from the VHE GRBs field, so I put some more emphasis/suggestions there, where is my field of expertise, but I kept in mind that this is a more general review, still I think there is a lack of/unbalanced mention to this very new and exciting developments.
Abstract:
Consider restructuring/splitting the first sentence, it's too long.
Intro:
L22. Their arrival times are unpredictable as is their arrival direction
L23 overcomes -> outshines (?)
L35 The goal
L39 on-board
L42 A citation around this moment might be good
L70 covered all the sky
L100 GRB histories (?)
L101 capable of establishing
L102 You have not defined "GRB durations" but in the figure 4 T90 appears You can define it in this part of the text "GRB prompt emission duration T90".
L110 a matter of debate
L113 points of view
L130 Despite some convictions...
L148 20-years aged
L152 "band" is repeated
L155: The sentence starting there is hard to read. I suggest splitting it in two or shortening it at least.
L160 WFCs not defined before. This sentence is also too long.
L192 put in italics "industrial phase A"
L199 Holland -> The Netherlands
L201 rephrase to remove "given" two times
L206 Remove Thus
L208 should -> could (also not sure what this sentence means)
L211 sure-> clear or secured
L256 Add proper citation to Wikipedia of Giuseppe Occhialini
L257 had a 3-month duration
L343 non-thermal process (this part needs a citation)
L360 ..was still relatively bright. Our collaborators...
Figure 20. Could you explain what the black and blue points are in the caption?
L641. I would suggest adding here the concept of the canonical light curve - arXiv:astro-ph/0508332, which is very much in hand with the work mentioned in this part and Fig 24
L672 About GRB130427A , one could add the citation to arXiv:1311.5245
Section 10. There is a lack of mension of the new GRB era (GeV-TeV GRBs) that I think it merits at least a paragraph somewhere around here. In connection with Sec 11.1 One of the still open issues is the emission mechanism of the VHE signals (Inverse Compton (SSC) vs Synchrotron), also worth mentioning briefly. Which I am not sure if its what you mention at around L771. If that's the case, then is not only 3 GRBs, but four (at least).
Comments on the Quality of English Language
English is ok, few hyphens and articles missing, some noted in my "main" feedback.
Reviewer 2 Report
Comments and Suggestions for Authors
** referee report **
The article titled "A Short History of the First 50 Years: from the GRB Discovery to the Afterglow and Multiwavelength Era" written by Filippo Frontera describes the history of GRBs from the GRB discovery to, mainly, the discovery of afterglows by BeppoSAX. Especially, because the author is the instrument PI of one of the BeppoSAX scientific instruments, uncovered stories regarding BeppoSAX are mentioned in very details, and I found this article very interesting. Here are my comments.
Main comments
1. The title of the article sounds like the paper is describing the history of a GRB up to the latest discoveries of GRBs. However, the article mainly focus on the history, discoveries and behind a scene stories about BeppoSAX. Therefore, I would like to request to change the title a bit so that a reader immediate notice that the article mainly focusing on the story of BeppoSAX.
2. section 6.1 (line 447-451): the spectral feastures seen in a GRB spectrum have been debated by several authors (e.g., Sako et al, ApJ, 623, 973-999). I personally think that those features are not conclusively accepted by the GRB community. I would like to suggest to tone down the section title and the sentence of this section.
3. section 6.2 (line 459-465): the column density in X-ray spectrum is strongly coupled by a photon index of a simple power-law in the case of GRB afterglow spectrum. And also only a few percent of Swift GRBs show a signs of a decreaseing NH which is not the majority of the sample. Again, I would suggest to tone down the section title and the sentence of this section.
4. line 566: Although it is written in the later part of the article, X-ray afterglow lightcurves observed in the Swift/XRT **do not** connect with a simple power-law decay index \beta. Thanks to the extremely rich Swift/XRT X-ray afterglow data, we all understand that the temporal properties of X-ray afterglow shows a very complex behavior. Please re-word the sentence to reflect our current understanding of the tempral properties of X-ray afterlow.
Minor comments
1. Line 7: comma is missing after "Venera."
2. Line 32: I personally think the word "spy" is not a good choice.
3. Line 47: According to my recollection, Vela satellites have changed the time resolution of the data. For example, Vela 1 was 32 s, Vela 3 was 0.5 s, Vela 4 was 1/8 s, and Vela 5/6 was 64 ms. It might be nice to mention about the time resolution improvements over the Vela satellites.
4. Line 228: "The GRBM proposal had also an international echo." The word "echo" is not clear to me.
5. Line 368: GRB970508 -> GRB 970508
6. Line 384: BeppoSAXwas -> BeppoSAX was
7. Line 391: Neil Gehrels Swift -> Neil Gehrels Swift Observatory
8. section 7.1: Please add the following references to the Swift scientific instruments:
BAT: Barthelmy et al. 2005, Space Sci. Rev., 120, 143
XRT: Burrows et al. 2005, Space Sci. Rev., 120, 165
UVOT: Roming et al. 2006, Space Sci. Rev., 120, 95
9. Line 683: Ruffini et al.[127] -> Ruffini et al. [127]
10. Line 689: after the event[128]. -> after the event [128].
Reviewer 3 Report
Comments and Suggestions for Authors
Dear the author,
Thank you for the excellent review. Here are minor comments only:
- l.4 long time => long-time
- l.20 case => cases
- l.24 research => researches
- l.24 (A little more general comment) For me it is a bit difficult to get the long sentence. I expected X-ing A, Y-ing B,,, but then normal nouns ("progress") followed without any verb (Z-ing).
Rather, it would be very helpful to have a paper outline in the usual format, i.e., Sec 1 is for XXX, etc. (Or, simply an 'itemize' table of contents, though I don't think it suits for this paper)
- l.33 spacecrafts
- l.187 l.199 Holland => Netherlands would be more appropriate
- l.188 on board of => onboard ?
- l.256 I would remove "(see Wikipedia)" as it does not convey any info. Maybe "for what" is needed like "see XXX for details"
- l.310 where t is the time since trigger
- l.425-6 However","
with respect to the => than in the...
remove "," after "shocks"
- l.434 favorite => favorable
- l.603 It was interesting "to see" (or know) the fact
- l.684 I would remove "(CO_{core})"
- l.769 "and thus it can be" or "and thus can it be"
- l.854 latter => last, or "the last, Vera Rubin Observatory, will..."
Best regards,
The referee
Comments on the Quality of English LanguageI don't have any other comment other than listed in the above.
